# Clinically distinct metabotypes of pediatric MASLD identified through unsupervised clustering of NASH CRN data

Helaina E. Huneault [1,2] ✉, Pradeep Tiwari[3,4], Zachery R. Jarrell [5], Matthew Ryan Smith [4,5], Chih-Yu Chen[6], Ana Ramirez Tovar[1,7], Cristian Sanchez-Torres[7], Scott Gillespie[8], Shasha Bai[8], Rodrigo M. Carrillo-Larco[3], Ajay K. Jain[9], Katherine P. Yates [10], Brent A. Neuschwander-Tetri [11], Jeffrey B. Schwimmer[12,13], Stavra A. Xanthakos[14], Jean P. Molleston[15], Cynthia A. Behling[16], Mark H. Fishbein[17], Terryl J. Hartman[1,18], Francisco J. Pasquel [19], Rishikesan Kamaleswaran[20,21], Dean P. Jones [5], Jean A. Welsh[1,7,22] & Miriam B. Vos[1,2,7,22,23]

Metabolic dysfunction-associated steatotic liver disease (MASLD) is the most common liver disease worldwide, yet treatment remains "one size fits all," despite phenotypic heterogeneity. We analyzed clinical and metabolomics data from 514 children (ages 5-18, 73% male) with biopsy-proven MASLD across three NASH Clinical Research Network studies. Unsupervised clustering of clinical data identified three distinct metabotypes: early-mild (49.4%, youngest, lowest lipids, liver enzymes, insulin resistance), cardiometabolic (36.8%, highest waist circumference, lipids, uric acid, SBP), and inflammatory-fibrotic (13.8%, highest liver enzymes, steatohepatitis, advanced fibrosis). Integrative network and pathway enrichment analyses revealed alterations in tryptophan metabolism within the inflammatory-fibrotic group, including elevated kynurenine pathway metabolites, which were significantly correlated with fibrosis stage. Branched-chain amino acid degradation, butanoate, and purine metabolism demonstrated greater enrichment in the cardiometabolic group. Here, we show that pediatric MASLD subtypes differ in clinical and metabolic features, providing a framework for targeted interventions, with validation needed in independent cohorts.

Metabolic dysfunction-associated steatotic liver disease (MASLD), previously known as nonalcoholic fatty liver disease (NAFLD), is the most prevalent liver disease in children and is a leading contributor to liver-related morbidity and mortality[1,2]. Once considered rare, the prevalence of pediatric MASLD has significantly increased and now affects an estimated 16.5% of U.S. adolescents[3], including 26% of those with obesity[4]. MASLD is characterized by the presence of hepatic steatosis ≥5%, along with at least one out of five specified cardiometabolic risk factors, in the absence of heavy alcohol use and other chronic liver diseases[1]. In children and adolescents, these cardiometabolic risk factors include increased body mass index (BMI), fasting glucose, blood pressure, triglycerides (TG), and low-density lipoprotein cholesterol (HDL). MASLD severity exists on a spectrum that ranges from simple steatosis to metabolic dysfunction-associated steatohepatitis (MASH), formerly known as nonalcoholic steatohepatitis (NASH), which can progress to fibrosis and cirrhosis,

predisposing one to hepatocellular carcinoma[1]. The current and future health burden of pediatric MASLD is immense, both for affected individuals and from a public health perspective. The disease presents a significant clinical challenge as it is associated with additional cardiovascular, metabolic, and psychological disorders[5,6]. Patient quality of life is also adversely affected, and there is an enormous economic burden due to healthcare utilization, which is likely to increase in adulthood[7]. Despite this burden, there are currently no FDA-approved medications or supplements for treating pediatric MASLD, and lifestyle modifications have demonstrated limited sustained success in reducing hepatic steatosis and serum alanine aminotransferase (ALT) levels, a key biomarker of liver injury and disease progression in MASLD[8-12].

Recent studies in adults have highlighted that MASLD is a heterogeneous condition with a broad spectrum of clinical presentations, pathogenesis, and disease severity[13-17]. This heterogeneity stems from various factors, including age, sex, hormonal status, genetics, gut microbiota composition, comorbidities, and certain exposures, including diet and physical activity[14]. Moreover, there is substantial inter-individual variation in response to current therapies for MASLD, which complicates treatment approaches[13,18,19]. Recent research in adults has identified sub-phenotypes of MASLD using clinical, genetic, and omics data[20-27]. These studies have uncovered differences in MASLD development and progression linked to genetic variants, lipid partitioning, body composition, and inflammation[28]. Given the heterogeneity observed in adults, it is plausible that sub-phenotypes also exist in children with MASLD. In line with the global shift towards precision medicine and public health, identifying MASLD phenotypes in youth could guide personalized interventions to prevent the onset and progression of hepatic steatosis. Unsupervised clustering analyses have been successfully used to identify subtypes in various diseases[29-31]. In this work, we identify three clinically distinct metabolic phenotypes (metabotypes) of pediatric MASLD, each characterized by unique metabolic and histological profiles, using unsupervised clustering and high-resolution metabolomics (HRM).

## Results

### NASH Clinical Research Network (CRN) cohort characteristics
Among the 514 children with biopsy-confirmed MASLD, ages ranged from 5 to 18 years, with a median age of 12 years. The cohort was predominantly male (73%) and of Hispanic ethnicity (74%), with 69% diagnosed with borderline or definite MASH. Notable differences were observed across the three NASH CRN study cohorts. The proportion of male participants included in our analysis was highest from the Treatment of NAFLD in Children (TONIC) study (85%) compared to the NAFLD Pediatric Database (DB1) (57%) and the NAFLD Pediatric Database 2 (DB2) ($p = 0.006$). The percentage of participants of Hispanic ethnicity also varied, with 77% in DB2, 65% in DB1, and 62% in TONIC ($p = 0.009$). Significant differences in liver enzyme levels were observed, with ALT and aspartate aminotransferase (AST) being highest in the TONIC and DB2 cohorts ($p < 0.001$). Glycated hemoglobin (HbA1c), BMI z-score, and NAFLD activity scores (NAS) also varied significantly across cohorts ($p < 0.05$). Please see Supplementary Data 1 for additional details.

### Clinically distinct metabotypes of pediatric MASLD
Our unsupervised clustering analysis revealed three clinically distinct metabotype clusters of pediatric MASLD (Table 1; Fig. 1). Based on the underlying metabolic phenotype, the clusters were categorized as follows: 1) early-mild (49.4% of participants), 2) cardiometabolic (36.8%), and 3) inflammatory-fibrotic (13.8%). The early-mild metabotype was significantly younger with lower levels of lipids (very-low-density lipoprotein cholesterol [VLDL], TG, low-density lipoprotein cholesterol [LDL]), liver enzymes (ALT, AST), and insulin resistance (Homeostasis Model Assessment of Insulin Resistance [HOMA2-IR])

(all $p < 0.001$). The cardiometabolic metabotype presented with the highest waist circumference (WC), VLDL, TG, and uric acid levels, with significantly elevated systolic blood pressure (SBP) ($p < 0.001$). Lastly, the inflammatory-fibrotic metabotype exhibited significantly higher levels of liver enzymes and had a greater percentage of participants with steatohepatitis and advanced fibrosis ($p < 0.001$). Interestingly, the early-mild and inflammatory-fibrotic metabotypes had higher percentages of Hispanic participants (81% and 82%, respectively) than the cardiometabolic metabotype (61%; $p < 0.001$). Furthermore, the early-mild metabotype had a higher percentage of participants with advanced fibrosis than cardiometabolic (17% vs. 8%, respectively; $p = 0.009$). Fig. S1 shows that the silhouette widths for the observed data consistently exceed those from the permuted datasets, indicating that the observed clusters are stronger and more meaningful than would be expected by chance. Additionally, as illustrated in Fig. S1, cluster differences were not primarily driven by sex, as male and female participants were represented across all clusters with no clear separation along the first two principal components.

### Metabolomic profiling of pediatric MASLD metabotypes
As described above, the primary goal of our HRM analysis was to examine differences in the metabolome across the three metabotypes. The hydrophilic interaction liquid chromatography positive mode (HILIC+) and C18 negative mode (C18-) principal component analysis (PCA) plots showed minimal separation among the early-mild, inflammatory-fibrotic, and cardiometabolic groups. However, the partial least squares discriminant analysis (PLS-DA) plots revealed clearer separation, with cardiometabolic displaying more distinct clustering, while early-mild and inflammatory-fibrotic still exhibited some overlap, suggesting the presence of unique metabolic profiles among the three metabotypes (Fig. S3 and S4). One-way analysis of variance (ANOVA) identified 248 significant features from HILIC+ mode and 178 from C18- mode (false discovery rate [FDR] $p < 0.05$: Supplementary Data 2 & 3). Among these, the most significant HILIC+ annotations included uric acid, which was highest in the cardiometabolic metabotype, as well as a phosphatidylcholine species (PC 36:2) and a sphingolipid species (SM 34:1), both elevated in the early-mild metabotype (all $p < 0.05$). In C18− mode, the top annotated metabolites included uric acid and related compounds involved in uric acid metabolism, such as 5-hydroxyisourate and urate, along with leucine/isoleucine. Uric acid, 5-hydroxyisourate, and urate levels were highest in the cardiometabolic metabotype ($p < 0.001$). Leucine/isoleucine was also significantly higher in the cardiometabolic group compared to the early-mild group ($p < 0.0001$) and showed a trend toward statistical significance compared to the inflammatory-fibrotic group ($p = 0.065$) (Fig. S3C and S4C).

Pathway enrichment analysis of the significant HILIC+ and C18-features identified by one-way ANOVA (Fig. 2; and Supplementary Data 4) revealed six metabolic pathways that were significantly altered across the three metabotypes. These pathways included tryptophan metabolism ($p < 0.001$), branched-chain amino acid (BCAA) degradation ($p = 0.001$), butanoate metabolism ($p = 0.004$), propanoate metabolism ($p = 0.008$), pantothenate and CoA biosynthesis ($p = 0.02$), and purine metabolism ($p = 0.02$). Group-wise distributions of the enriched features in each pathway are illustrated in Fig. S5-S10 using heatmaps and violin plots. Notably, several metabolites within the tryptophan metabolism pathway, including tryptophan, serotonin, kynurenine, and indole-3-acetaldehyde, as well as pantothenate and CoA biosynthesis (aspartate and serine), were elevated in the inflammatory-fibrotic metabotype compared to cardiometabolic and early-mild (all $p < 0.05$; Fig. S5 and S6). In contrast, purine metabolism, BCAA degradation, and butanoate metabolism showed higher levels of metabolites in the cardiometabolic metabotype relative to the other two groups, while several metabolites of propanoate metabolism were higher in the early-mild metabotype ($p < 0.05$; Fig. S7−S10).

**Table 1 | Clinical, Histologic, and Demographic Characteristics of Children and Adolescents with MASLD stratified by metabotype**

| Characteristic | Early-mild n = 254 | Cardiometabolic n = 189 | Inflammatory-fibrotic n = 71 | p-value |
|---|---|---|---|---|
| Study | | | | $8.00 \times 10^{-4}$ |
| DB1 | 11 (4.3%) | 12 (6.3%) | 0 (0%) | |
| DB2 | 213 (84%) | 130 (69%) | 55 (77%) | |
| TONIC | 30 (12%) | 47 (25%) | 16 (23%) | |
| Age (yrs.) | 11 (9, 12) | 15 (13, 16) | 12 (11, 13) | $1.85 \times 10^{-37}$ |
| Male (n,%) | 174 (69%) | 147 (78%) | 56 (79%) | 0.048 |
| Hispanic race/ethnicity (n,%) | 206 (81%) | 116 (61%) | 58 (82%) | $4.90 \times 10^{-6}$ |
| BMI (kg/m2) | 28.9 (26.0, 31.6) | 35.2 (32.1, 39.0) | 32.0 (28.3, 35.2) | $8.64 \times 10^{-32}$ |
| BMI Z-score | 2.2 (1.9, 2.6) | 2.4 (2.1, 2.7) | 2.3 (2.0, 2.8) | $4.01 \times 10^{-4}$ |
| BMI %ile | 98.6 (96.9, 99.6) | 99.3 (98.4, 99.8) | 99.2 (98.0, 99.8) | $5.47 \times 10^{-5}$ |
| WC (cm) | 96 (87, 104) | 114 (104, 123) | 106 (97, 113) | $2.09 \times 10^{-38}$ |
| ALT (U/L) | 77 (54, 111) | 79 (56, 105) | 256 (219, 300) | $2.20 \times 10^{-39}$ |
| AST (U/L) | 48 (35, 62) | 45 (35, 60) | 136 (117, 164) | $2.89 \times 10^{-39}$ |
| GGT (U/L) | 31 (23, 45) | 38 (27, 54) | 65 (49, 106) | $6.48 \times 10^{-22}$ |
| Glucose (mg/dL) | 87 (81, 94) | 88 (81, 93) | 88 (83, 97) | 0.257 |
| Insulin (uU/L) | 21 (14, 30) | 35 (23, 53) | 39 (22, 58) | $8.90 \times 10^{-19}$ |
| HbA1c (%) | 5.3 (5.1, 5.6) | 5.3 (5.1, 5.7) | 5.4 (5.2, 5.6) | 0.458 |
| HOMA-B | 195 (146, 265) | 286 (204, 389) | 301 (197, 386) | $1.03 \times 10^{-16}$ |
| HOMA-S | 39 (27, 57) | 23 (16, 35) | 21 (14, 36) | $7.58 \times 10^{-19}$ |
| HOMA2-IR | 2.6 (1.8, 3.7) | 4.4 (2.9, 6.3) | 4.8 (2.8, 7.0) | $7.58 \times 10^{-19}$ |
| TG (mg/dL) | 110 (77, 143) | 170 (129, 228) | 143 (111, 215) | $4.98 \times 10^{-23}$ |
| TC (mg/dL) | 149 (129, 173) | 179 (159, 206) | 182 (151, 210) | $4.32 \times 10^{-19}$ |
| VLDL-c (mg/dL) | 22 (15, 29) | 34 (26, 46) | 29 (22, 43) | $4.98 \times 10^{-23}$ |
| HDL-c (mg/dL) | 40 (34, 45) | 36 (32, 42) | 40 (33, 45) | $6.83 \times 10^{-4}$ |
| LDL-c (mg/dL) | 87 (69, 107) | 105 (91, 127) | 105 (80, 130) | $4.20 \times 10^{-13}$ |
| TG:HDL ratio | 2.7 (1.8, 3.7) | 4.6 (3.2, 7.0) | 3.8 (2.7, 6.0) | $1.29 \times 10^{-22}$ |
| Alk phos (U/L) | 263 (214, 311) | 165 (103, 261) | 251 (206, 316) | $5.49 \times 10^{-16}$ |
| Bilirubin (mg/dL) | 0.4 (0.3, 0.6) | 0.5 (0.4, 0.7) | 0.5 (0.3, 0.6) | $1.38 \times 10^{-4}$ |
| Creatinine (mg/dL) | 0.5 (0.4, 0.6) | 0.6 (0.5, 0.7) | 0.5 (0.4, 0.6) | $1.65 \times 10^{-24}$ |
| Albumin (g/dL) | 4.5 (4.3, 4.7) | 4.6 (4.3, 4.7) | 4.6 (4.4, 4.8) | 0.005 |
| Uric acid (mg/dL) | 5.2 (4.5, 6.0) | 6.8 (5.9, 7.7) | 5.8 (4.9, 6.6) | $2.96 \times 10^{-32}$ |
| Platelets (cells/uL) | 293,000 (258,000, 338,000) | 285,000 (252,000, 322,000) | 274,000 (243,000, 332,000) | 0.243 |
| SBP (mm Hg) | 114 (107, 123) | 127 (118, 135) | 122 (112, 129) | $5.94 \times 10^{-21}$ |
| DBP (mm Hg) | 65 (59, 70) | 71 (65, 77) | 69 (63, 74) | $2.06 \times 10^{-11}$ |
| Steatosis score (n,%) | | | | 0.635 |
| 0 | 2 (0.8%) | 2 (1.1%) | 1 (1.4%) | |
| 1 | 69 (27%) | 57 (30%) | 24 (34%) | |
| 2 | 81 (32%) | 69 (37%) | 20 (28%) | |
| 3 | 102 (40%) | 61 (32%) | 26 (37%) | |
| Lobular inflammation (n,%) | | | | $1.89 \times 10^{-6}$ |
| 0 | 2 (0.8%) | 0 (0%) | 0 (0%) | |
| 1 | 170 (67%) | 124 (66%) | 25 (35%) | |
| 2 | 76 (30%) | 58 (31%) | 36 (51%) | |
| 3 | 6 (2.4%) | 7 (3.7%) | 10 (14%) | |
| Ballooning score (n,%) | | | | $2.74 \times 10^{-15}$ |
| 0 | 185 (73%) | 115 (61%) | 18 (25%) | |
| 1 | 55 (22%) | 53 (28%) | 26 (37%) | |
| 2 | 14 (5.5%) | 21 (11%) | 27 (38%) | |
| Fibrosis stage (n,%) | | | | $7.65 \times 10^{-11}$ |
| 0 | 83 (33%) | 77 (41%) | 9 (13%) | |
| 1 | 100 (39%) | 66 (35%) | 12 (17%) | |
| 2 | 29 (11%) | 31 (16%) | 25 (35%) | |
| 3 | 39 (15%) | 13 (6.9%) | 23 (32%) | |
| 4 | 3 (1.2%) | 2 (1.1%) | 2 (2.8%) | |

**Table 1 (continued) | Clinical, Histologic, and Demographic Characteristics of Children and Adolescents with MASLD stratified by metabotype**

| Characteristic | Early-mild<br>$n = 254$ | Cardiometabolic<br>$n = 189$ | Inflammatory-fibrotic<br>$n = 71$ | p-value |
|---|---|---|---|---|
| NAS (n,%) | | | | **$2.42 \times 10^{-6}$** |
| 1 | 2 (0.8%) | 1 (0.5%) | 0 (0%) | |
| 2 | 47 (19%) | 33 (17%) | 2 (2.8%) | |
| 3 | 60 (24%) | 48 (25%) | 10 (14%) | |
| 4 | 69 (27%) | 53 (28%) | 20 (28%) | |
| 5 | 52 (20%) | 24 (13%) | 15 (21%) | |
| 6 | 21 (8.3%) | 20 (11%) | 12 (17%) | |
| 7 | 2 (0.8%) | 9 (4.8%) | 8 (11%) | |
| 8 | 1 (0.4%) | 1 (0.5%) | 4 (5.6%) | |
| Advanced fibrosis (n,%) | 42 (17%) | 15 (7.9%) | 25 (35%) | $5.68 \times 10^{-7}$ |
| MASH (n,%) | | | | $5.21 \times 10^{-24}$ |
| *No MASH* | 84 (33%) | 72 (38%) | 3 (4%) | |
| *Borderline Zone 3 pattern* | 23 (9%) | 39 (21%) | 15 (21%) | |
| *Borderline Zone 1 pattern* | 113 (44%) | 31 (16%) | 9 (13%) | |
| *Definite MASH* | 34 (13%) | 47 (25%) | 44 (62%) | |
| MASH (any) (n,%) | 170 (67%) | 117 (62%) | 68 (96%) | **$5.61 \times 10^{-7}$** |

Participant characteristics of the 514 children and adolescents with biopsy-confirmed MASLD are displayed across the three metabotype clusters: Early-mild, Cardiometabolic, and Inflammatory-fibrotic. Significant differences across clusters ($p < 0.05$) were assessed using two-sided Kruskal–Wallis tests for continuous variables and two-sided chi-squared or Fisher's exact tests, as appropriate, for categorical variables. Statistically significant $p$ values ($p < 0.05$) are shown in bold.

Cluster characteristics were reported as median (interquartile range) for continuous variables and frequencies (percentages) for categorical variables. Comparisons across the clusters were performed using the Kruskal–Wallis test for continuous variables and either the chi-squared test or Fisher's exact test for categorical variables, as appropriate. Statistically significant differences ($p < 0.05$) were highlighted in bold. *Abbreviations: DB1* Pediatric Database 1, *DB2* Pediatric Database 2, *TONIC* Treatment of Nonalcoholic fatty liver disease in children, *BMI* body mass index, *%ile* percentile, *WC* waist circumference, *ALT* alanine aminotransferase, *AST* aspartate aminotransferase, *GGT* gamma-glutamyl transferase, *HbA1c* hemoglobin a1c, *HOMA-B* homeostatic model assessment of beta cell function, *HOMA-S* homeostatic model assessment of insulin sensitivity, *HOMA2-IR* updated homeostatic model assessment of insulin resistance, *TG* triglycerides, *TC* total cholesterol, *VLDL-c* very low-density lipoprotein cholesterol, *HDL-c* high-density lipoprotein cholesterol, *LDL-c* low-density lipoprotein cholesterol, *TG:HDL* ratio triglycerides to high-density lipoprotein ratio, *Alk phos* alkaline phosphatase, *SBP* systolic blood pressure, *DBP* diastolic blood pressure, *NAS* NAFLD activity score, *MASH* metabolic dysfunction-associated steatohepatitis.

Among the significantly enriched pathways, several contained at least one Schymanski Level Confidence (SLC) Level 1 identification from our internal reference library, supporting the confidence of these results[32]. Three pathways, propanoate metabolism, BCAA degradation, and pantothenate and CoA biosynthesis, contained features that could not be independently confirmed by secondary annotation methods and were therefore reported as SLC Level 5. These results should be interpreted cautiously but may represent true biological signals and provide important hypotheses for future validation.

### Network analysis of clinical & metabolomic features across pediatric MASLD metabotypes

Figure 3 shows the integrative networks of the clinical and metabolic features for the three metabotypes. VLDL and TG had the highest centrality scores across all networks, underscoring their prominent roles within each group (Supplementary Data 5). The inflammatory-fibrotic network also showed high centrality for clinical markers associated with inflammation and metabolic dysfunction, such as ALT, AST, fibrosis stage, and HOMA2-IR, reflecting their central roles within this network. Similarly, the early-mild network included clinical variables such as ALT, AST, fibrosis stage, and HOMA2-IR within the same community, suggesting related pathways or functions among these variables in both networks. In contrast, the cardiometabolic network was predominantly lipid-focused, with LDL and WC positioned alongside VLDL and TG, highlighting a central orientation around lipid and adiposity markers. These orientations and centrality differences across the networks underscore the distinct metabolic heterogeneity characterizing each of the three metabotypes.

Comparing the early-mild and cardiometabolic metabotype networks, we identified three significantly enriched pathways based on metabolic features differing in centrality: tryptophan metabolism, pantothenate and CoA biosynthesis, and BCAA degradation ($p < 0.05$; Fig. 3; and Supplementary Data 6). Pairwise comparisons showed that the majority of these metabolites, including 3-methyl-2-oxobutanoic acid, valine, and 4-methyl-2-oxopentanoate, were significantly elevated in the cardiometabolic metabotype versus early-mild ($p < 0.05$; Fig. S11). Similar pathway alterations were observed in the comparison between the early-mild and inflammatory-fibrotic metabotype networks; however, glycine, serine, and threonine metabolism was also significantly enriched, with features in these pathways such as tryptophan, kynurenine, aspartate, and hydroxyproline, notably higher in the inflammatory-fibrotic metabotype ($p < 0.05$; Fig. 3 and S12). Lastly, when comparing the inflammatory-fibrotic and cardiometabolic networks, tryptophan metabolism, BCAA degradation, pantothenate, and CoA biosynthesis, as well as glycine, serine, and threonine metabolism, were significantly enriched. Metabolic features from tryptophan and glycine, serine, and threonine metabolism were significantly higher in the inflammatory-fibrotic metabotype ($p < 0.05$), while those from BCAA degradation, including leucine/isoleucine, were elevated in the cardiometabolic metabotype (Fig. 3 and S13). Given the consistent alteration of metabolites in the tryptophan pathway across metabotypes, we further examined their associations with clinical variables in the network analysis. Notably, tryptophan, indole-3-acetaldehyde, serotonin, 5-methoxyindoleacetate, 5-phenyl-1,3-oxazinane-2,4-dione, and kynurenine showed moderate to strong positive correlations with ALT, AST, and fibrosis stage (Supplementary Data 7).

### Correlations between metabolic features and fibrosis stage

Among the 3,758 mass-to-charge *(m/z)* features from HILIC+ mode and 3520 from C18- mode, 351 and 249 features, respectively, showed significant associations with the fibrosis stage ($p < 0.05$; Supplementary Data 8). As shown in Fig. 4, the top metabolites correlated with fibrosis stage were 4-hydroxy-5-methyl-3(2H)-thiophenone, a sulfur-containing thiophene derivative potentially involved in oxidative stress pathways[33,34] ($r = 0.23$, $p = 1.42 \times 10^{-7}$; HILIC+ mode), and an unknown feature (437.1615_66.9; $r = -0.23$, $p = 3.03 \times 10^{-7}$; C18- mode).

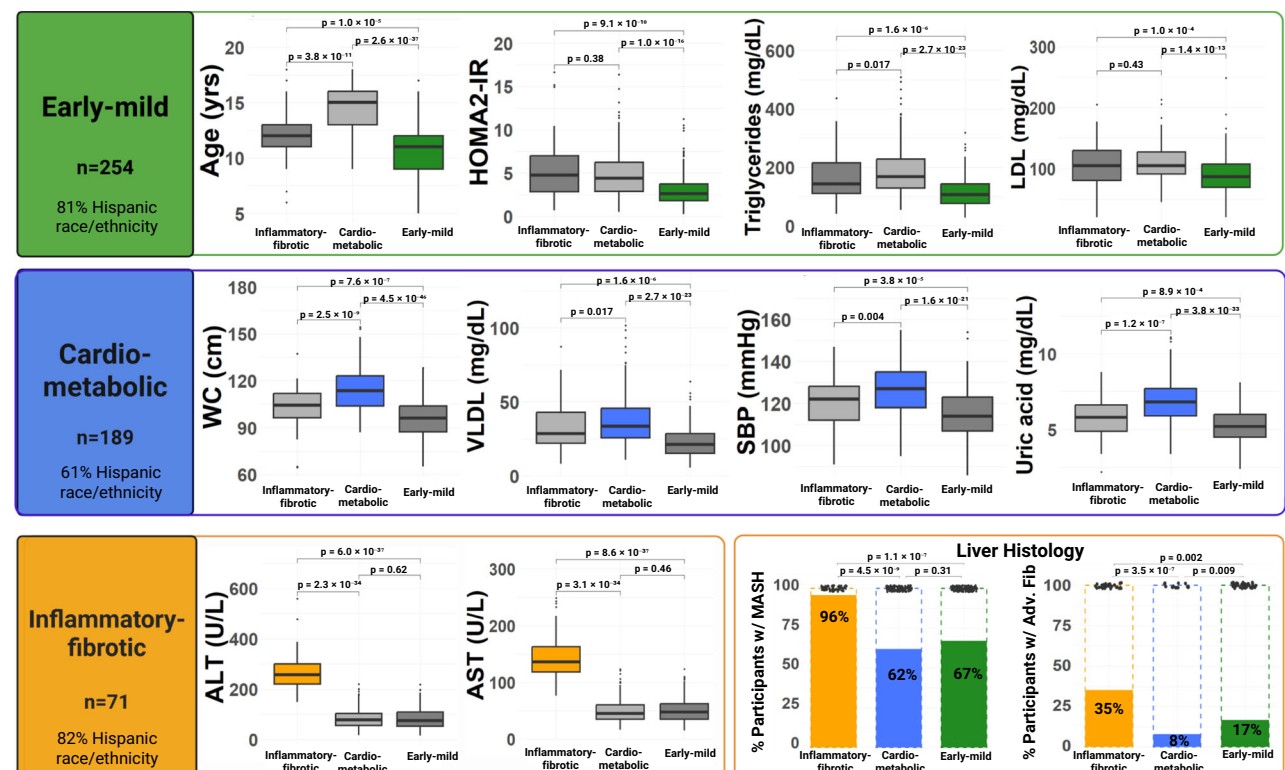

**Fig. 1 | Clinically Distinct Metabotypes of Pediatric MASLD.** Boxplots illustrating the distribution of the ten clinical predictors among 514 children and adolescents with MASLD, stratified by metabotype cluster: Inflammatory fibrotic (n = 71, orange), Cardiometabolic (n = 189, blue), and Early-mild (n = 254, green). For all boxplots, the center line represents the median, the box bounds represent the interquartile range (IQR; 25th and 75th percentiles), whiskers extend to 1.5× IQR, and points outside the whiskers denote outliers. Bar charts show the proportion of participants with MASH and advanced fibrosis based on liver histology. Pairwise comparisons were conducted using two-sided Mann-Whitney U tests for continuous variables and two-sided chi-squared or Fisher's exact tests for categorical variables. Statistical significance was defined as p < 0.05. Created in BioRender. Huneault, H. (https://BioRender.com/fqq5drs).

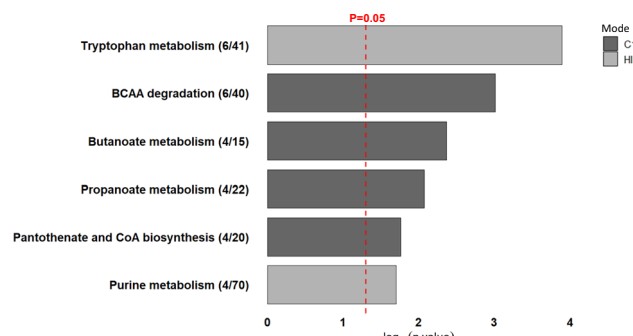

**Fig. 2 | Pathway enrichment analysis of metabolomic features.** Pathway analysis results using Mummichog v2.0, based on significant metabolite features identified in one-way ANOVA (284 from HILIC+ mode and 178 from C18- mode). The pathways shown were significantly enriched (p < 0.05) as determined by permutation testing, with each pathway containing at least three overlapping features. The red dotted vertical line marks the threshold at p = 0.05. The bars are color-coded by ionization mode, with HILIC+ in light gray and C18- in dark gray. Numbers in parentheses denote the number of significant metabolic features (numerator) out of the number of detected metabolic features within that pathway (denominator).

Additionally, several tryptophan-related metabolites, such as indole, tryptophan, and kynurenine, were positively associated with fibrosis, suggesting a link between tryptophan metabolism and fibrosis severity. Lipid-related metabolites, including phosphatidylethanolamine (PE), phosphatidylcholine (PC), and phosphatidic acid (PA) species, also showed positive correlations with fibrosis stage. Furthermore,

polycyclic aromatic hydrocarbons (PAHs), such as pyrene and 2-aminonaphthalene, as well as nucleotide metabolism-related features, including deoxyinosine and xanthine, and bile acid-related metabolites such as chenodeoxycholic acid derivatives, were also positively correlated with fibrosis.

## Discussion

MASLD encompasses a wide range of clinical presentations, underlying mechanisms, and disease severity. Although MASLD heterogeneity has been explored in adults, pediatric research remains limited, focusing primarily on histology-defined subtypes[35,36]. This study investigated clinically relevant phenotypes of pediatric MASLD without relying exclusively on histological assessment. Using an unsupervised clustering approach, we identified three distinct metabotypes, each with unique clinical and metabolomic profiles. Our findings highlight the heterogeneity of MASLD in youth and the potential to improve health outcomes for children with MASLD by developing tailored interventions for each metabotype.

As illustrated in Fig. 5, we hypothesize that the early-mild metabotype represents a younger, less severe phenotype with mixed features of the other two metabotypes, which may progress into one of them over time. Given the inflammatory-fibrotic metabotype's elevated liver enzymes, liver inflammation and fibrosis, lower lipid levels, and a higher representation of participants of Hispanic background, this subtype likely carries an increased risk of specific genetic polymorphisms, such as the patatin-like phospholipase domain-containing protein 3 (PNPLA3) variant. This polymorphism, which is prevalent among individuals of Hispanic descent, is known to impair VLDL secretion by limiting triglyceride mobilization, potentially elevating the risk of hepatic complications in this group[37]. Unfortunately,

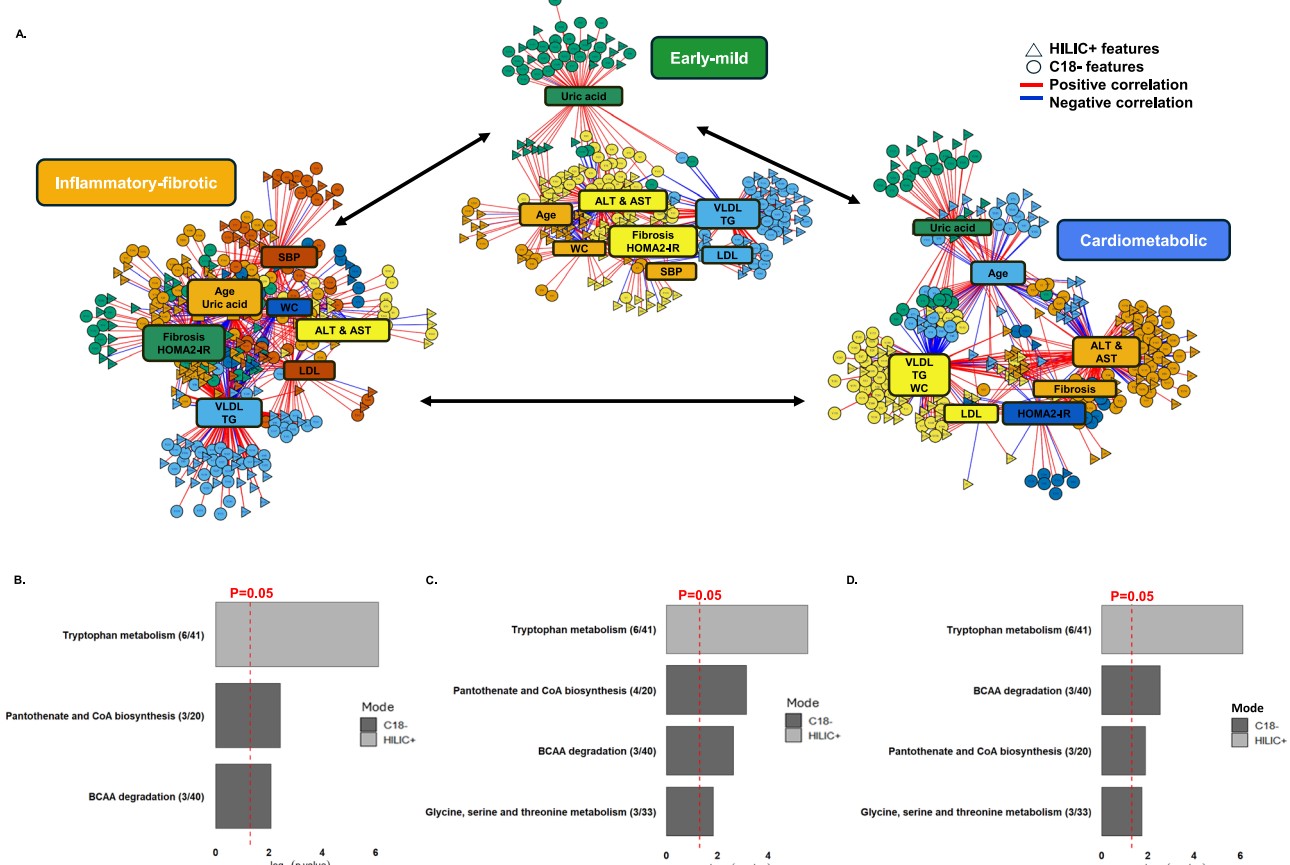

**Fig. 3 | Integrative network and pathway analysis across pediatric MASLD metabotypes.** Integrative network & pathway analysis of clinical biomarkers, fibrosis score, and serum metabolomics data in children and adolescents from the combined NASH CRN cohort stratified by metabotype groups after outlier removal ($n = 490$): Inflammatory-fibrotic (IF, $n = 67$), Early-mild (EM, $n = 248$), and Cardiometabolic (CM, $n = 175$). **A** Within each metabotype network, communities of highly connected metabolite features and clinical variables, represented by distinct colors, were identified using the multilevel community detection algorithm. Associations between metabolite features and clinical variables are illustrated by edges, with red lines indicating positive correlations and blue lines indicating negative correlations. Associations were determined by partial least squares canonical mode, with significance thresholds of $|r| > 0.30$ and $p < 0.05$. HILIC+ and C18-

features are represented by triangles and circles, respectively. **B** Pair-wise pathway analysis results for EM vs. CM, **C** EM vs. IF, and (**D**) IF vs. CM based on metabolite features that differed in centrality by $\geq 0.025$ between the metabotype networks. The pathways shown were significantly enriched ($p < 0.05$) as determined by permutation testing using the mummichog algorithm, with each pathway containing at least three overlapping features. All statistical tests were two-sided, and $p$ values were not adjusted for multiple comparisons. The red dotted vertical line marks $p = 0.05$. Bars are color coded by ionization mode, with HILIC+ in light gray and C18- in dark gray. Numbers in parentheses denote the number of significant metabolic features (numerator) out of the total detected features in the pathway (denominator).

genotyping data is not currently available for this cohort. Conversely, the cardiometabolic metabotype, characterized by significant dyslipidemia, abdominal obesity, elevated uric acid levels, and SBP, but lower liver inflammation, fibrosis, and fewer participants of Hispanic ethnicity, likely does not have impaired VLDL secretion, increasing the risk of extrahepatic complications, particularly cardiovascular disease (CVD). The higher proportion of advanced fibrosis observed in the early-mild group underscores the potential for fibrosis to develop early in life, even in the absence of overt metabolic dysfunction, and highlights the need for longitudinal studies to better understand progression pathways in pediatric MASLD[35].

Recent studies investigating MASLD subtypes in adults have revealed findings comparable to our pediatric metabotypes. Carrillo-Larco et al. used NHANES III data to identify three phenotypes: Average, Lipid-liver, and Anthro-SBP-glucose[38]. Among these, the Anthro-SBP-glucose phenotype exhibited the highest risk of all-cause mortality and closely aligns with our cardiometabolic metabotype. Similarly, Yi et al. and Ye et al. identified distinct MASLD clusters, including subtypes characterized by severe insulin resistance and poor survival outcomes[39,40]. While these findings overlap with our metabotypes,

differences in clustering criteria and MASLD progression between adults and children highlight the importance of pediatric-specific research. Notably, we did not observe significant differences in glucose levels across groups, likely reflecting variations in insulin resistance associated with puberty in youth[41]. Moreover, Romeo et al. identified two MASLD subtypes through genome-wide association analysis and polygenic risk scores: a liver-specific subtype linked to genetic variants associated with hepatic triglyceride retention (PNPLA3 and transmembrane 6 superfamily member 2 [TM6SF2]) that presented with more aggressive liver disease but reduced cardiovascular risk, and a systemic subtype associated with increased CVD risk[42]. These findings support our hypothesis that the inflammatory-fibrotic metabotype involves impaired VLDL secretion and hepatic complications, particularly in Hispanic populations where the PNPLA3 variant is more prevalent[43]. Conversely, our cardiometabolic metabotype mirrors the systemic subtype, characterized by dyslipidemia, hypertension, and an increased risk of extrahepatic complications and CVD.

A recent longitudinal study published in this journal by Raverdy et al.[27] expanded on Romeo's work by performing a clustering analysis of adults with obesity, identifying two MASLD subtypes: a liver-specific

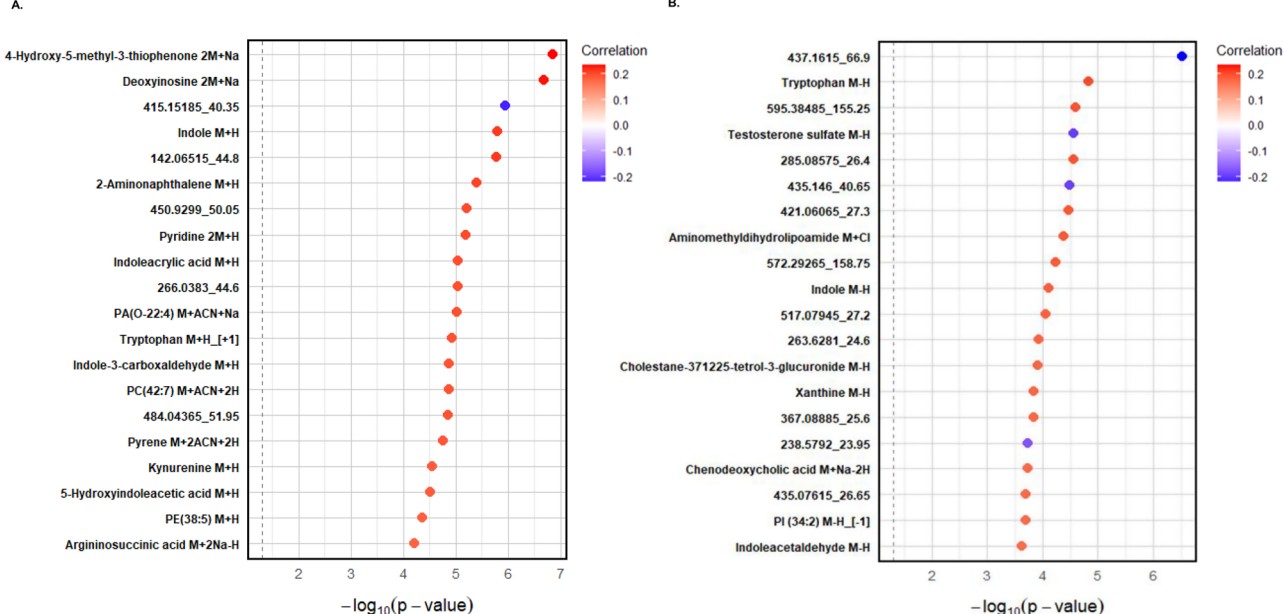

**Fig. 4 | Metabolite features associated with fibrosis stage in pediatric MASLD.** Top 20 HILIC+ (**A**) and C18− (**B**) metabolite features associated with fibrosis among NASH CRN study participants after outlier removal (*n* = 490). Correlations are presented on a color scale from −1 (blue) to 1 (red). The dashed line indicates a *p* value cutoff of <0.05. Spearman rank correlations were calculated between each metabolite feature and fibrosis score using the rcorr function in the Hmisc package. All statistical tests were two sided, and *p* values were not adjusted for multiple comparisons. Metabolite features were annotated using the internal reference library and xMSannotator. Please see Table S8 for correlations, *p* values, adducts, and annotation confidence. Abbreviations: PA phosphatidic acid, PC phosphatidylcholine, PE phosphatidylethanolamine, PI phosphatidylinositol. O- denotes an ether-linked species.

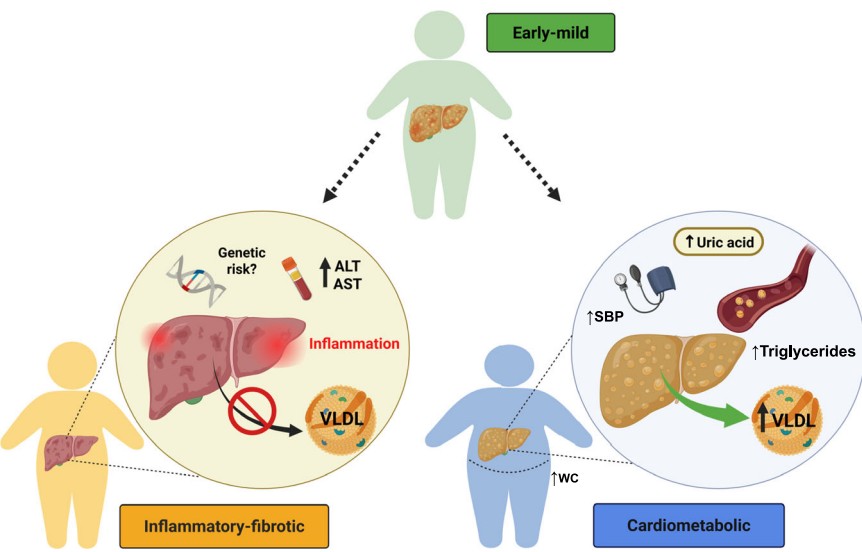

**Fig. 5 | Clinically distinct metabotypes of pediatric MASLD.** The Early-mild (EM) metabotype may represent a younger, less severe, mixed phenotype of the Inflammatory-fibrotic (IF) and Cardiometabolic (CM) groups, potentially differentiating into one of these metabotypes over time. Given the IF metabotypes elevated liver enzymes, liver inflammation, lower lipid levels, and higher representation of participants of Hispanic descent, this subtype likely carries an increased risk for specific genetic polymorphisms, such as the PNPLA3 variant. This polymorphism, which is prevalent among individuals of Hispanic descent, is known to impair VLDL secretion by limiting triglyceride mobilization, potentially elevating the risk of hepatic complications in this group. Conversely, the CM metabotype, characterized by significant dyslipidemia, abdominal obesity, elevated uric acid levels, and SBP, but lower liver inflammation and fewer participants from Hispanic backgrounds, likely does not have impaired VLDL secretion, increasing the risk of extrahepatic complications, particularly cardiovascular disease (CVD). Created in BioRender. Huneault, H. (https://BioRender.com/pwitjny).

subtype and a cardiometabolic subtype. Remarkably, our findings closely parallel theirs despite the lack of longitudinal data in our study. Specifically, our inflammatory-fibrotic metabotype aligns with the liver-specific subtype, and we observed striking similarities between the pediatric and adult cardiometabolic subtypes. Notably, they did not find significant differences in chronic liver disease risk between subtypes, likely due to the more advanced disease seen in their adult population. Furthermore, while their study identified differences in diabetes risk, we did not observe this in our analysis, which may be attributed to the progressive decline in pancreatic function that occurs

in adults with prolonged insulin resistance[44]. These findings highlight the importance of longitudinal pediatric studies to understand how MASLD subtypes in childhood influence disease trajectories and outcomes in adulthood.

In our exploratory metabolomics analysis, we identified significant alterations in several metabolic pathways, including tryptophan metabolism, BCAA degradation, purine metabolism, butanoate metabolism, and pantothenate and CoA biosynthesis, which were differentially represented across the three metabotypes.

Metabolites of tryptophan metabolism, including tryptophan, kynurenine, and serotonin, were elevated specifically in the inflammatory-fibrotic metabotype, suggesting a key role for this pathway in the progression of liver disease within this subgroup. The kynurenine pathway, regulated by the enzyme indoleamine 2,3-dioxygenase (IDO), is upregulated by pro-inflammatory cytokines and linked to increased inflammation and fibrosis in MASLD[45,46]. Elevated kynurenine levels in the inflammatory-fibrotic group correlated with liver injury markers, such as ALT, AST, and fibrosis stage, highlighting the role of tryptophan metabolism in driving inflammatory and fibrotic processes. These findings align with prior studies demonstrating that tryptophan and serotonin exacerbate hepatic lipid accumulation and inflammation, reinforcing the importance of targeting this pathway in therapeutic interventions[47,48]. Tryptophan catabolism is also upregulated in obesity and is associated with systemic low-grade chronic inflammation[49,50]. While enrichment of tryptophan metabolism in the inflammatory-fibrotic group likely reflects fibrosis-related processes, its elevated presence in the cardiometabolic group may reflect underlying metabolic inflammation. Fibrosis progression in pediatric MASLD likely results from the combined effects of altered tryptophan metabolism, lipid metabolism, and inflammation. Future studies are needed to clarify the specific contributions of tryptophan metabolism to fibrosis development and disease progression in this population.

The pantothenate and CoA biosynthesis pathway was also higher in the inflammatory-fibrotic metabotype compared to the other two groups. Alterations in pantothenate and CoA biosynthesis are linked with mitochondrial dysfunction and hepatocyte apoptosis, which may contribute to the metabolic stresses observed in the inflammatory-fibrotic group[51,52]. Furthermore, our network analysis revealed elevated metabolites involved in glycine, serine, and threonine metabolism in the inflammatory-fibrotic metabotype, notably hydroxyproline, which plays a key role in hepatic fibrosis[53,54].

In the cardiometabolic metabotype, purine metabolites, particularly uric acid, were elevated. Elevated uric acid levels are linked to obesity, insulin resistance, and endothelial dysfunction in MASLD[55]. Uric acid is also associated with excess fructose consumption, which bypasses glycolytic regulation and depletes ATP, leading to AMP accumulation and subsequent degradation into uric acid[56]. Although dietary intake data were unavailable, the elevated uric acid in the cardiometabolic group may partly reflect excessive fructose consumption. While fructose intake might also influence other metabotypes, its effect in the cardiometabolic group could contribute to the increased CVD risk associated with this subtype.

In our network analysis, metabolites of BCAA degradation, including leucine/isoleucine, were consistently elevated in the cardiometabolic metabotype compared to inflammatory-fibrotic and early-mild. Elevated circulating BCAA levels are associated with obesity, insulin resistance, and an increased fatty liver index[57]. Insulin normally promotes BCAA uptake and degradation, but insulin resistance disrupts this process, reducing BCAA breakdown and increasing circulating levels[58]. Additionally, the branched-chain α-keto acid dehydrogenase (BCKDH) enzyme, which catalyzes a key step in BCAA catabolism, is often downregulated in metabolic disorders, including MASLD[59]. This impairment in the BCAA degradation pathway may be particularly relevant in the cardiometabolic metabotype.

Moreover, butanoate metabolism showed significant alterations in the cardiometabolic metabotype relative to early-mild and inflammatory-fibrotic. This pathway is linked to microbial-derived metabolites that may act locally within liver tissue[60]. A recent review highlighted butanoate metabolism as one of the most dysregulated lipidome-related signatures in MASLD, suggesting that microbial dysbiosis may contribute to the cardiometabolic metabotype[61]. However, the potential harmful or beneficial impacts of butanoate metabolism on human health remain under investigation. Interestingly, Raverdy et al. also identified elevated gut microbiota-associated metabolites in their cardiometabolic cluster, which reflect patterns similar to those observed in our pediatric cardiometabolic metabotype[27]. Lastly, metabolites involved in propanoate metabolism were upregulated in the early-mild metabotype compared to the inflammatory-fibrotic and cardiometabolic groups. This pathway is also associated with microbial dysbiosis and may indicate early changes occurring in MASLD development[62]. However, fewer alterations were observed in this pathway, possibly due to the less severe metabolic dysfunction in the early-mild metabotype.

Advances in precision healthcare are redefining the management of metabolic disorders by developing treatments tailored to patients' unique biochemical and physiological profiles[63]. Although further research is needed to validate our findings in an independent cohort, the significant heterogeneity observed among children with MASLD in our study highlights the need for clinical applications that address the specific metabolic dysfunctions of each metabotype. For the inflammatory-fibrotic metabotype, tailored lifestyle interventions could focus on reducing inflammation and protecting liver health. This may involve an anti-inflammatory diet, avoiding high-fructose products, including sugar-sweetened beverages, treatment with anti-fibrotic medications, and potentially therapeutic targets of tryptophan and kynurenine metabolism. In contrast, managing the cardiometabolic metabotype might prioritize cardiovascular health and the cultivation of a beneficial gut-microbial composition. Suggested interventions include a heart-healthy, fiber-rich diet that limits sugar-sweetened beverages and calorie-dense foods combined with regular physical activity. Early cardiovascular disease surveillance and pharmacologic options such as metformin, statins, and anti-obesity medications, including glucagon-like peptide-1 receptor agonists (GLP-1 RAs), could also be considered. Lastly, clinical applications for the early-mild metabotype could focus on early identification and prevention, aiming to mitigate progression toward more severe metabolic dysfunction.

In this study, unsupervised machine learning was used to identify clinically relevant subtypes of pediatric MASLD, offering insights into disease heterogeneity beyond histology-based approaches. A key strength of our study was the use of data from NASH CRN cohorts, a well-characterized pediatric population in which all participants had biopsy-confirmed MASLD, providing detailed clinical and histological data. Additionally, the application of high-resolution metabolomics allowed us to explore metabolic pathways associated with each metabotype, offering valuable insights into potential mechanisms driving metabolic dysfunction across the groups. The cross-sectional design of our study limits our ability to make causal inferences or assess longitudinal outcomes such as CVD or advanced liver disease. Moreover, the lack of a control group and the predominance of males of Hispanic descent in our sample may also limit the generalizability of our findings. Additional limitations include the lack of genetic, gut-microbial, and dietary data, which could provide further insight into the metabolic differences observed. While our untargeted metabolomics approach offers broad coverage of the metabolome, it is not optimized for targeted lipidomics. Future studies employing dedicated lipidomics platforms are needed to better characterize lipid-driven mechanisms underlying pediatric MASLD heterogeneity. Moreover, further research could explore the relationship between specific metabolite features, clinical variables, and fibrosis severity in

pediatric MASLD to identify potential biomarkers of fibrosis risk and refine our understanding of disease progression. Finally, our pathway enrichment analysis was exploratory and not intended to establish definitive pathways or biomarkers; targeted studies are required to validate these findings.

In conclusion, this study identified three distinct metabotypes of pediatric MASLD, each with unique clinical and metabolic profiles, highlighting significant heterogeneity within this population. Our findings underscore the potential for precision healthcare approaches to tailor interventions to specific metabolic dysfunctions in MASLD subtypes. Given the exploratory nature of the metabolomics analysis, further longitudinal and targeted studies are needed to validate these findings and explore strategies that may improve outcomes for children with MASLD.

## Methods

All participants included in this study provided written informed consent and assent for their samples and data to be used for future research. Written informed consent was obtained from a parent or legal guardian for all participants, regardless of age. Participants aged 13-17 years provided written adolescent assent, and those under 13 years provided written child assent. Research was conducted in accordance with both the Declarations of Helsinki and Istanbul, and the study protocol was approved by the institutional review boards at Emory University and Children's Healthcare of Atlanta (Atlanta, GA; STUDY00001715).

This retrospective cross-sectional study employed a two-step framework (Fig. 6). First, a data-driven clustering analysis was performed using anthropometric and clinical data from youth enrolled in

NASH CRN studies to identify metabotypes of pediatric MASLD. Second, an exploratory high-resolution metabolomics analysis of participant serum samples was conducted to examine differences in the metabolome among the identified metabotypes.

### NASH CRN cohort

The sample population included a combined cohort of children and adolescents aged 5–18 years with liver biopsy-proven MASLD enrolled in one of three National Institute for Diabetes and Digestive and Kidney Diseases (NIDDK)-funded NASH CRN studies: the Treatment of NAFLD in Children (TONIC) trial ($n = 93$) [NCT00063635], the NAFLD Pediatric Database (DB1) study ($n = 23$), and the NAFLD Pediatric Database 2 (DB2) study ($n = 398$) [NCT01061684].

The NAFLD Pediatric Database studies (DB1 and DB2)[64] were observational analyses of patients two years and older with MASLD, and TONIC was a phase III, masked, randomized, placebo-controlled trial of metformin or vitamin E in children ages 8–17 years with MASLD[65]. Inclusion and exclusion criteria were similar among the three studies[64,66]. Briefly, eligible participants were ≤18 years of age with MASLD confirmed by liver biopsy. Exclusion criteria included any clinical or histologic evidence of alcoholic liver disease, evidence of other causes of chronic liver disease, greater than one month of total parenteral nutrition six months before the liver biopsy, short bowel syndrome, biliopancreatic diversion, bariatric surgery, and known HIV infection. Participants in the TONIC study were also required to have a baseline ALT value of ≥60 U/L, and additional exclusion criteria were the diagnosis of diabetes, metabolic acidosis or renal dysfunction, and the use of anti-diabetic or anti-MASLD drugs.

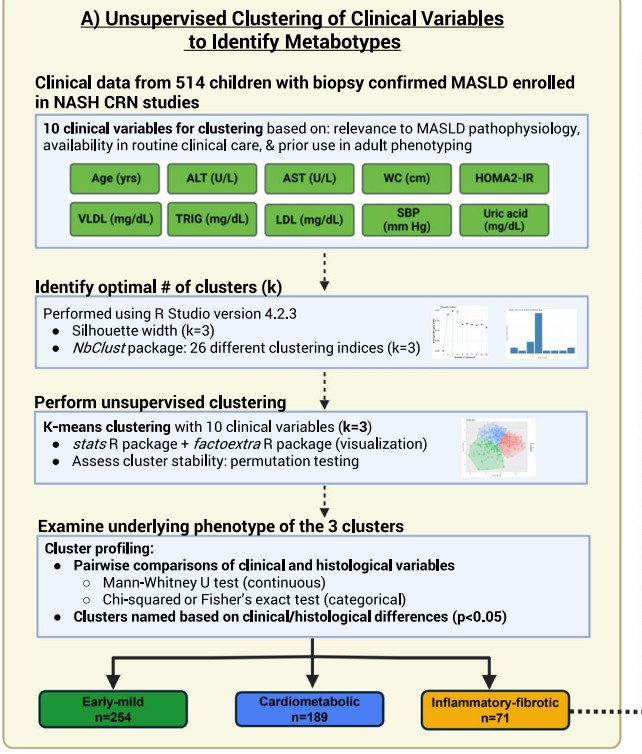

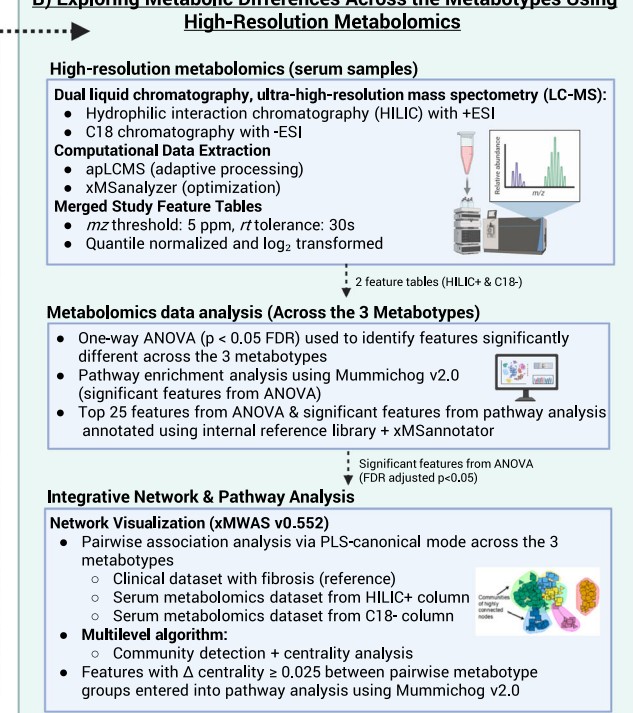

**Fig. 6 | Two-step framework for identifying metabotypes and exploring metabolic differences in pediatric MASLD. A** Unsupervised clustering of clinical variables from 514 children enrolled in NASH CRN studies with biopsy-confirmed MASLD identified distinct metabotypes using *k*-means clustering. **B** High-resolution metabolomics was performed on stored serum samples, including feature extraction (apLCMS, xMSanalyzer), quantile normalization, and log2 transformation. Features that differed significantly across metabotypes ($p < 0.05$, FDR-adjusted)

were identified using one-way ANOVA, entered into untargeted pathway analysis (Mummichog v2.0), and annotated using an internal reference library (tier 1) and xMSannotator. Metabolomics features were integrated with clinical biomarkers and fibrosis stage in xMWAS for network and pathway analysis. Features differing in centrality scores by ≥0.025 between metabotype networks were further explored in pathway enrichment analysis. Created in BioRender. Huneault, H. (https://BioRender.com/hzkj0ka).

For the present study, children were included if they met the current diagnostic criteria for MASLD confirmed by liver biopsy (hepatic steatosis ≥5%) along with at least one marker of cardiometabolic dysfunction[1] and had serum samples processed for high-resolution metabolomics at the Emory Clinical Biomarkers Laboratory. The initial eligible sample included 556 children and adolescents with available clinical and metabolomics data. A small percentage of missing data (0.34%) was imputed using the median for clinical variables. Outliers, defined as participants with any clinical variable greater than five standard deviations from the mean, were excluded, resulting in 515 participants. During this process, most participants with an HbA1c ≥ 6.5% were excluded; however, one participant remained in the dataset. To maintain consistency and avoid bias from a single value, this participant was also excluded, yielding a final sample of 514 children and adolescents for analysis (Fig. S14).

## Clinical and laboratory assessments

Demographic data for the three studies were collected through structured interviews and questionnaires. Height, weight, and waist circumference measurements were measured in duplicate, with participants standing in light clothing without shoes. Body mass index (BMI) was calculated by dividing weight (kg) by height (m) squared. BMI percentiles were determined based on age and sex using the Centers for Disease Control and Prevention (CDC) criteria[67].

Fasting blood samples were collected from all participants after an overnight fast (minimum of 8 h or more) and processed for plasma and serum within 2 h[64,65]. Laboratory assays included in the present study were ALT, AST, alkaline phosphatase, gamma-glutamyl transferase (GGT), platelet count, fasting glucose, insulin, HbA1c, bilirubin, and uric acid levels. Additionally, a lipid panel including fasting TG, total cholesterol, LDL, and HDL was included. VLDL was calculated by dividing triglyceride levels by five, consistent with established methods[68]. The updated homeostasis model assessment of insulin resistance (HOMA2-IR) was calculated using the online HOMA calculator version 2.2.3[69]. Systolic and diastolic blood pressure were assessed using standard procedures[70]. Additional fasting samples from all study participants were stored at −80 °C at the NIDDK Central repository for future use. Please see our Supplementary Materials for a table summarizing participant baseline clinical and demographic characteristics stratified by study (Supplementary Data 1).

## Histological evaluation

For each study, biopsies were centrally assessed by the NASH CRN Pathology Committee as a group for consensus scoring according to the system by Kleiner et al[71]. Assessments included the NAFLD activity score (NAS) on a scale of 0-8, with higher scores indicating more severe disease; the components of this measure include steatosis (0-3), lobular inflammation (0–3), and hepatocellular ballooning (0-2). Biopsies were also evaluated for the presence or absence of MASH and the stage of fibrosis, which was classified as stage 0, stage 1a (mild perisinusoidal), stage 1b (moderate perisinusoidal), stage 1c (portal/periportal fibrosis only), stage 2 (zone 3 and periportal), stage 3 (bridging fibrosis), and stage 4 (cirrhosis). For analysis, participants with fibrosis stages 1a, 1b, and 1c were grouped into a single category, and fibrosis stage ≥3 was considered advanced fibrosis. The diagnosis of MASH was divided into four categories: no MASH, borderline zone 1, borderline zone 3, and definite MASH[72]. The borderline and definite MASH categories were also combined into a single group for analysis based on studies in adults indicating a similar risk of fibrosis progression in borderline MASH patients compared to those with definite MASH[73].

## Unsupervised clustering analysis using clinical data

The clinical variables used for unsupervised clustering were selected based on their association with MASLD, ease of availability in clinical practice, and prior use in adult MASLD phenotyping studies. These ten variables included age (years), ALT (U/L), AST (U/L), WC (cm), VLDL (mg/dL), LDL (mg/dL), TG (mg/dL), HOMA2-IR, uric acid (mg/dL), and SBP (mm Hg), which informed the cluster composition and helped characterize the underlying phenotypes. All variables were scaled to a mean of 0 and a standard deviation of 1 before performing $k$-means clustering to ensure comparability across different scales. The optimal number of clusters ($k$) was determined using several approaches, including the silhouette width, and the NbClust package in R Studio version 4.2.3 (R Foundation for Statistical Computing, Vienna, Austria)[74], which provides a comprehensive evaluation of 26 different clustering indices to recommend the optimal number of clusters[75]. Based on these methods, the majority indicated three clusters as optimal (Fig. S15 and S16). $K$-means clustering was then performed with a k value of three, using the $k$-means function from the stats package[76] in R Studio (maximum iterations: 1000, nstart: 25), and results were visualized with the factoextra R package[77]. The $k$-means algorithm partitions the data by minimizing within-cluster variance using a centroid-based distance metric for efficient unsupervised clustering[78]. Permutation testing was also conducted to assess cluster stability by generating 100 randomized datasets, each created by independently permuting the values within each variable (Fig. S1). $K$-means clustering was applied to each permuted dataset, and the average silhouette width was calculated for cluster numbers ranging from $k = 2$ to 10. Since sex is a categorical variable and could not be included in $k$-means clustering, principal component analysis (PCA) was used to reduce the data to the first two principal components, enabling cluster visualizations to highlight differences by sex (Fig. S1).

## Cluster profiling

All variables were evaluated for normality, and descriptive statistics were calculated to summarize the characteristics of the three clusters. Categorical variables were reported as counts and frequencies, while continuous variables were presented as medians with interquartile ranges (IQR). Pairwise comparisons between clusters were performed using the Mann-Whitney U test for continuous variables and either the chi-squared test or Fisher's exact test, as appropriate, for categorical variables. Key results were illustrated using boxplots and bar charts. The clusters were labeled based on the clinical and histological variables that showed statistically significant differences and best characterized their underlying phenotypes. Statistical analyses were conducted in RStudio version 4.2.3, with statistical significance set at $p < 0.05$. OpenAI's ChatGPT version 4 was used to refine R code and assist with minor language improvements, including grammar and readability, during manuscript preparation. The final manuscript, including all scientific content and conclusions, was thoroughly reviewed and validated by the authors.

## High-resolution metabolomics (HRM)

Participant samples from each study were safely transferred from the NIDDK Central Repository to Emory University, where they were stored at -80 °C. While serum samples from the DB2 cohort were processed in 2020 and those from the TONIC and DB1 cohorts in 2022, all analyses were performed at the Emory Clinical Biomarkers Laboratory using established liquid chromatography-mass spectrometry (LC-MS) methods, including identical instrumentation, protocols, and internal quality control procedures to minimize potential batch-related variability[32,79]. Additionally, serum samples were randomized prior to extraction and LC-MS acquisition to minimize potential bias from instrument drift and batch position effects. All samples were extracted with acetonitrile (ACN) plus an internal standard and analyzed in batches of 40 using a dual chromatography platform. The platform consisted of hydrophilic interaction liquid chromatography (HILIC) with positive electrospray ionization (ESI) and C18 reversed-phase chromatography with negative ESI. Serum samples from the

TONIC and DB1 studies were analyzed using an Orbitrap Fusion mass spectrometer (ThermoFisher Scientific, San Jose, CA), while samples from the DB2 study were analyzed on an HFQE mass spectrometer (ThermoFisher Scientific, San Jose, CA), both operated at 120,000 resolution in full-scan mode. Each batch of samples included pooled human plasma (QStd-3) at the beginning, middle, and end, along with NIST 1950-calibrated reference pooled human plasma preceding and following each batch. This dual chromatography platform provides broad coverage of the metabolome, including ~500 endogenous, dietary, and environmental chemicals[80], with identities confirmed by co-elution and MS/MS fragmentation spectra matching authentic standards[32]. Raw data files were extracted to generate feature tables using apLCMS[81] and xMSanalyzer[82], which facilitate adaptive alignment of untargeted metabolomics data and merging of features across multiple extraction parameters. Spectral feature intensities were corrected for batch effects using ComBat[83] and filtered for the coefficient of variations less than 75%. Triplicate analyses were tested for replicability using Pearson correlation (r > 0.7), and triplicates were median summarized prior to further analysis. The resulting data file from the TONIC and DB1 studies consisted of 18,826 HILIC/ + ESI and 18,024 C18/-ESI chemical features defined by an accurate mass to charge ($m/z$), retention time (RT), and ion abundance, while the resulting HILIC/ + ESI and C18/-ESI data files from the DB2 study consisted of 12,614 and 8468 $m/z$ features, respectively. The two separate HILIC+ and C18- datasets were matched and merged based on a $m/z$ cutoff of 5 ppm and an RT cutoff of 30 seconds using R Studio software version 4.2.3. The resulting 3758 m/z-matched features from HILIC + and 3520 from C18- were standardized across the studies, quantile normalized, and log2-transformed using the online software MetaboAnalyst version 6.0 prior to downstream analysis. A summary of our analytical workflow is illustrated in Fig. 6.

## Metabolomics data analysis and feature annotation

Principal component analysis (PCA) and partial least squares discriminant analysis (PLS-DA) were used to examine overall patterns and group separation among the three metabotypes. PCA was initially applied to identify potential outliers based on the Hotelling's $T^2$ statistic with a 99% confidence interval across the first five principal components[84]. Outlier removal, performed in R Studio version 4.2.3, led to the exclusion of 24 samples (4.7% of the dataset), possibly due to pre-analytical variations among samples, such as participant location or sample freeze times, for which specific data were not available. One-way ANOVA was performed in MetaboAnalyst version 6.0, both with and without the identified outliers, to confirm that their removal did not alter the overall results. Significant features identified by ANOVA, following FDR correction, were further analyzed using Tukey's Honest Significant Difference (HSD) post hoc test for pairwise comparisons. The top 25 differentiating HILIC+ and C18-features were first annotated using an internal reference library and confirmed by matching ion dissociation patterns and retention times with authentic standards, which were reported as Schymanski Level 1[85]. The remaining features were computationally annotated using xMSannotator, which performs accurate mass matching to common positive and negative mode adducts in the Human Metabolome Database (HMDB) with an $m/z$ tolerance of ± 5 parts per million (ppm) and a retention time tolerance of 10 seconds[86]. Annotation confidence was reported using the Schymanski Level Confidence (SLC) framework, which ranges from Level 1 (confirmed structure with reference standard) to Level 5 (exact mass of interest only)[32].

Pathway enrichment was conducted with Mummichog version 2.0[87] in MetaboAnalyst 6.0[88]. Significant features identified through one-way ANOVA with FDR-adjusted $p$ values < 0.05 were used as input. Enrichment was evaluated via permutation testing using all quality-filtered m/z features as the background. Pathways were considered enriched if the permutation-derived $p$ value was <0.05 and at least

three significant features overlapped with known metabolites in the pathway. Pathway membership was defined according to the Kyoto Encyclopedia of Genes and Genomes (KEGG) library. Analyses were performed with the following parameters: mass tolerance 5 ppm; RT in seconds; primary ions enforced; and inclusion of pathways/metabolite sets containing ≥3 entries. Significant features identified through pathway analysis were also annotated as described above. The pathway results include ratios that indicate the number of significant features mapped to each pathway (numerator) out of the total number of pathway-relevant features (denominator). Group-wise distributions of the significant features were examined to interpret pathway differences between groups and visualized using the R packages pheatmap[89] for heatmaps and ggplot2[90] for violin plots. Pathway enrichment was performed as an exploratory approach to investigate potential biological differences between clinically defined metabotypes, recognizing that further targeted studies will be necessary to confirm these findings. The datasets used as input for pathway analysis via MetaboAnalyst are provided in Supplementary Data 9 and **10**.

## Integrative network analysis and pathway enrichment analysis

The 284 significant metabolic features from HILIC+ and 178 from C18-, identified through one-way ANOVA, were integrated with the clinical dataset using the data integration software xMWAS (v.0.552)[91]. To reduce skewness, the following clinical variables were log-transformed prior to data integration: ALT, AST, TG, VLDL, LDL, and HOMA2-IR. The fibrosis stage was also included in the clinical dataset due to its significant variation across the three clusters, and was square root transformed to address zero values. In xMWAS, partial least squares (PLS) canonical mode was used to conduct pairwise association analysis between the metabolomics and clinical datasets for each metabotype. This approach identifies bidirectional associations between clinical variables and metabolic features[92]. The clinical dataset was entered as the reference, and associations were selected based on association scores ($|r| > 0.3$) and significance ($p < 0.05$ by Student's t-test). Network visualization and community detection were performed for each metabotype based on the resulting association matrix using a multilevel algorithm designed to identify hierarchical community structures consisting of tightly connected nodes representing metabolic features and clinical variables[93]. The eigenvector centrality method was utilized to identify nodes exhibiting changes in their topological characteristics between networks. Nodes with higher centrality scores have more connections and are considered more influential within the network[94,95]. An absolute difference in centrality score (Δ centrality) of 0.025 was selected to identify metabolic features with meaningful changes in centrality across networks. This threshold was selected to equilibrate loading of selected metabolites for pathway analysis across all comparisons while limiting inclusion of minimal centrality shifts. The features with a Δ centrality of ≥0.025 between network pairs were entered into pathway enrichment analysis using Mummichog version 2.0[87] via MetaboAnalyst 6.0[88]. Pathways were considered significantly enriched based on $p < 0.05$ in permutation-based testing and an overlap size ≥3 $m/z$ features.

## Correlation analysis of metabolic features with fibrosis stage

To identify metabolic features significantly correlated with the fibrosis stage, we used the rcorr function from the Hmisc package[96] in RStudio to calculate Spearman correlation coefficients and $p$ values for metabolites in both HILIC+ and C18- modes. The fibrosis stage was included as a continuous variable and square root transformed to address zero values and reduce skewness. The top 20 significant feature correlations ($p < 0.05$) were annotated using an internal reference library or xMSannotator, as described above, and visualized in a Manhattan plot, with $p$ values transformed to negative log10 to highlight significance levels.

**Reporting summary**

Further information on research design is available in the Nature Portfolio Reporting Summary linked to this article.

## Data availability

Clinical and metabolomics data from the NAFLD Pediatric Database 2 (DB2) NASH CRN study are available through the Human Health Exposure Analysis Resource (HHEAR) Data Center (Project ID: 2017-1593) and the Metabolomics Workbench (Study ID: ST001428). Clinical data from the Treatment of NAFLD in Children (TONIC) trial and the NAFLD Pediatric Database (DB1) NASH CRN studies are available through the NIDDK Central Repository. Metabolomics datasets generated from the TONIC and DB1 studies are available under controlled access due to NASH CRN data use agreements and participant privacy protections, and may be requested through the NIDDK Central Repository (https://repository.niddk.nih.gov/pages/for_requestors). All processed data supporting the findings of this study, including metabolite feature tables, statistical outputs, pathway enrichment results, and figure source data, are provided in the accompanying Source Data files. Raw, identifiable clinical data cannot be shared due to NIDDK data use policies and participant privacy regulations. Source data for Table 1 and Supplementary Data 1 are not publicly available to protect participant privacy due to the presence of indirect identifiers, but may be obtained through request to the corresponding author for purposes of interpreting, verifying, and extending the research described in this article, subject to institutional approvals and data use agreements. Requests will be reviewed within 30 days. Source data are provided with this paper.

## Code availability

The code used to perform the analyses in this study is publicly available at: (https://github.com/HHuneault/NASH_CRN_pediatric_metabotypes_R). This repository includes scripts for the unsupervised clustering, metabolomics analysis, and associated figures described in the manuscript. DOI: Huneault, H. Code for *Clinically Distinct Metabotypes of Pediatric MASLD*. Zenodo. https://doi.org/10.5281/zenodo.17807234 (2025).

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

## Acknowledgements

This work was supported by the National Center for Advancing Translational Sciences of the National Institutes of Health under Award Numbers UL1TR002378 (MBV), UL1TR000454 (MBV), and UL1TR000448 (MBV); by NIH grants R01DK125701 (MBV), K23DK080953 (MBV), and NR019083 (MBV); by the US Department of Veterans Affairs Career Development Award 5IK2BX005913-02 (MRS); and by the NIDDK NASH CRN cooperative agreements U01DK061731 (MBV), U01DK061718 (MBV), U01DK061730(MBV), and U24DK061730 (MBV). Additional support was provided through the Human Health Exposure Analysis Resource (HHEAR) program under Award Numbers R01DK131136 (MBV) and R21AI169487 (MBV). The content is solely the responsibility of the authors and does not necessarily represent the official views of the National Institutes of Health or the U.S. Department of Veterans Affairs. We gratefully acknowledge the National Institute of Diabetes and Digestive and Kidney Diseases (NIDDK) for supporting the NASH Clinical Research Network (NASH CRN) and its collaboration with the Human Health Exposure Analysis Resource (HHEAR), developed by the National Institute of Environmental Health Sciences (NIEHS). We thank the NASH CRN investigators and the Ancillary Studies Committee for access to clinical samples and data from the NAFLD Database, NAFLD Pediatric Database 2 (NCT01061684), and the TONIC trial (NCT00063635). Biospecimens reported here were supplied by the NIDDK Central Repository. This manuscript was not prepared in collaboration with the NIDDK Central Repository and does not necessarily reflect its opinions or official views. We also thank Ken Liu, Mary Nellis, James Zhan, Joshua Preston, ViLinh Tran, William Crandall, and Jaclyn Weinberg of the Emory Clinical Biomarkers Laboratory for their technical support. Finally, we extend our deepest appreciation to the NASH CRN study participants and their families for their invaluable contributions.

## Author contributions

H.E.H. performed the statistical and metabolomics analysis, and P.T., S.G., S.B., R.M.C.L., Z.R.J., M.R.S., and C.Y.C. provided guidance. H.E.H. wrote the manuscript, and P.T., Z.R.J., C.Y.C., and M.B.V. provided guidance. P.T., Z.R.J., M.R.S., C.Y.C., A.R.T., C.S.T., S.G., S.B., R.M.C.L., A.K.J., K.P.Y., B.A.N.T., J.B.S., S.A.X., J.P.M., C.Y.B., M.H.F., T.J.H., F.J.P., R.K., D.P.J., J.A.W., and M.B.V. reviewed the final manuscript and provided feedback.

## Competing interests

MBV serves as a consultant to Boehringer Ingelheim, Novo Nordisk, Eli Lilly, Intercept, Takeda, and Alberio. She has stock or stock options in Thiogenesis and Tern Pharmaceuticals. Her institution has received research grants (or in-kind research services) from Target Real World Evidence, Quest, Labcorp, and Sonic Incytes Medical Corp. JPM has research grant funding from Gilead, Abbvie, Albireo, and Mirum. BANT: Advisor or consultant: Abbvie, Akero, Aldeyra, Aligos, Arrowhead, Corcept, Galectin, GSK, Hepion, HistoIndex, Madrigal, Merck, Mirum, Pfizer, Sagimet, Senseion; Stock options: HepGene, HeptaBio; Institutional research grants: Madrigal. SAX: Institutional research grants: Target Real World Evidence. The remaining authors declare no competing interests.

## Additional information

[1]Nutrition & Health Sciences Program, Laney Graduate School, Emory University, Atlanta, GA, USA. [2]Department of Pediatrics and Human Development, Michigan State University, College of Human Medicine, Grand Rapids, MI, USA. [3]Hubert Department of Global Health, Rollins School of Public Health, Emory University, Atlanta, GA, USA. [4]VA Healthcare System of Atlanta, Decatur, GA, USA. [5]Division of Pulmonary, Allergy, Critical Care, and Sleep Medicine, Department of Medicine, Emory University School of Medicine, Atlanta, GA, USA. [6]Emory Integrated Metabolomics and Lipidomics Core, School of Medicine, Emory University, Atlanta, GA, USA. [7]Department of Pediatrics, Division of Gastroenterology, Hepatology, and Nutrition, Emory University, Atlanta, GA, USA. [8]Pediatric Biostatistics Core, Department of Pediatrics, School of Medicine, Emory University, Atlanta, GA, USA. [9]Department of Pediatrics, Saint Louis University School of Medicine, Saint Louis, MO, USA. [10]Department of Epidemiology, Johns Hopkins Bloomberg School of Public Health, Baltimore, Maryland, USA. [11]Department of Internal Medicine, Saint Louis University, St. Louis, Missouri, USA. [12]Department of Gastroenterology, Rady Children's Hospital San Diego, San Diego, CA, USA. [13]Department of Pediatrics, School of Medicine, University of California, San Diego, La Jolla, CA, USA. [14]Division of Gastroenterology, Hepatology and Nutrition, Cincinnati Children's Hospital Medical Center, Department of Pediatrics, University of Cincinnati College of Medicine, Cincinnati, OH, USA. [15]Department of Pediatrics, Indiana University/Riley Hospital for Children, Indianapolis, IN, USA. [16]Sharp Memorial Hospital, San Diego, California, USA. [17]Department of Pediatrics, Feinberg School of Medicine at Northwestern University, Chicago, IL, USA. [18]Department of Epidemiology, Rollins School of Public Health and Winship Cancer Institute, Emory University, Atlanta, GA, USA. [19]Division of Endocrinology, Department of Medicine, Emory University School of Medicine, Atlanta, GA, USA. [20]Department of Surgery, Duke University School of Medicine, Durham, NC, USA. [21]Department of Anesthesiology, Duke University School of Medicine, Durham, NC, USA. [22]Children's Healthcare of Atlanta, Atlanta, GA, USA. [23]Helen DeVos Children's Hospital, Corewell Health, Grand Rapids, MI, USA. ✉e-mail: hhuneau@emory.edu

