## [Transparent Peer Review file · Nature Communications]

Clinically distinct metabotypes of pediatric MASLD identified through unsupervised clustering of NASH CRN data

Corresponding Author: Ms Helaina Huneault

Version 0:

Reviewer comments:

Reviewer #1

(Remarks to the Author)

The study investigates the Metabolic Dysfunction-Associated Steatotic Liver Disease (MASLD) in 3 pediatric cohorts assembled into one dataset containing demographic data, clinical targeted measurements and untargeted metabolomics features. The authors present the analytical chemistry method for metabolic feature discovery and the data analysis pipeline. In the first step of the data analysis, clinically distinct metabotypes were formed with unsupervised clustering based on the 10 manually selected features. Following that, the comparative analysis of metabolomics data was performed to identify the most distinct metabolic features of each cluster. Some features were annotated and used in pathway enrichment analysis.

The authors present important work on pediatric cohorts in a clear and comprehensive way. The reviewer is not a wet lab specialist and therefore cannot comment on the specifics of the LC-MS method. Thus, the comments are focused on the data analysis methods.

1. The central claim of the paper is the existence of the three distinct metabotypes, which directs the follow up statistical and pathway enrichment analysis. This analysis is performed based on the 10 preselected variables. While the unsupervised methods used for clustering and arriving at the conclusion about the existence of exactly 3 metabotypes are solid, the reviewer has concerns about the possible bias introduced by the choice of variables. In the Discussion section, the authors make connections between the findings of the presented study and the existing literature. However, the 10 variables are also preselected on the basis of "prior use in adult MASLD phenotyping studies", which makes them more likely to return similar results to the existing studies. The reviewer suspects that the results of the unsupervised clustering might significantly change when all the clinical and demographic variables are included. The results of the clustering that are based on all the variables must be included and compared to the presented metabotypes. Otherwise, the more detailed justification for not including the variables must be described.

2. Do the same metabotypes also occur when the unsupervised clustering of untargeted metabolomics features is performed? Do some other clusters form if only the metabolomics is measured and clinical variables are not included? How do they relate to the metabotypes formed by the clinical variables and the previous studies?

3. No participants from DB1 ended up in the inflammatory-fibrotic metabotype. Do the authors suspect that this due to chance or due to the selection bias of the participants in DB1?

4. To make an analysis on the Figure 4 even more explainable, the reviewer suggests including some potential cofactors (age, clinical variables) into a linear regression alongside with the most correlated metabolic features to predict the levels of fibrosis. This would allow to list out some possible cofactors and establish associations in the datasets.

5. The reviewer suggests that the Supplementary Tables are provided in the Excel/CSV format for easy access.

(Remarks on code availability)

The code is provided alongside with installation instructions.
The reviewer tested one file and it was executing as expected.

Reviewer #2

(Remarks to the Author)

In this study the authors used a very well characterized cohort of 514 children with biopsy-proven MASLD to identify clinically distinct metabolic phenotypes of paediatric MASLD using unsupervised clustering and high-resolution metabolomics. This work with the aim to uncover distinct metabolic phenotypes in children with MASLD is very timely and important and a significant step in paediatric liver disease. In the first step, they used clinical and histological parameters and identified 3 unique clusters. This part is done very well. Although validation using external cohort is required. The second part related to metabolomics is more concerning. 3 cohorts were done in two groups possibly in two different labs and 2 years apart, hence the initial number of features are quite different and then no randomization in analysis which is quite important. The methodology is also less suitable for analysing lipids which are important in MASH and important potential biomarkers were reported previously in adult MASH patients. In addition, the results are not well-described. They reported 284 significant features but it is not clear if these are the metabolites that differentiate the 3 identified clusters or did they performed unsupervised analysis independently? Are these fully annotated using standards (tier 1) or these are only features or mixed? Then without discussing specificity of these metabolites, they moved on with performing pathway analysis using Mummichog or MetaboAnalyst which are ok for quick visualization but not reliable for such an important differentiation. I really don't think this type of analysis is relevant for this work and it offers any meaningful outcome. I suggest to remove it! This study has a lot of potential with a very important cohort. It would be great to use the same thorough evaluation of the metabolites as it was done for the clinical and histological parameters.

Minor comment: Please use the recently approved term MASH/MASLD across the manuscript. Currently both MASH and NASH/NAFLD is used across the text.

(Remarks on code availability)

Reviewer #3

(Remarks to the Author)

Thank you for asking me to review this manuscript. This was a very interesting to read, and relevant to the field, piece of work.

Noteworthy results:

The work presented looks to identify distinct phenotypes/metabotypes of paediatric MASLD by analysing clinical and metabolomic data from children with biopsy proven MASLD who had previously participated in 3 NASH CRN studies. (The inclusion of biopsy proven MASLD is a strength of this work).

With the use of unsupervised clustering 3 distinct metabotypes were identified: the early-mild, the cardiometabolic and the inflammatory-fibrotic, with differences in clinical, laboratory and metabolic parameters between the groups. In addition six metabolic pathways were shown to be significantly altered between the three metabotypes.

Significance of work for the field:

This work would be of significance in the field, as it is very important to understand the different 'types' of MASLD in children from a disease pathogenesis point of view, from a prognostic point of view and of course from a management/ treatment point of view. Even more so as the prevalence of paediatric MASLD is rapidly increasing world wide. The work compares well to the relevant literature in adults with MASLD and the authors analyse this adequately in the discussion.

The work does support the conclusions and claims. The authors do highlight the limitations of the study, mainly the lack of a control group and the predominance of Hispanic males in the participants. This does limit generalisability of the conclusions, but the work still provides valuable insight that would be very helpful in the design of future studies including longitudinal research.

In terms of data analysis, interpretation and conclusions, there are a few points that may strengthen the manuscript should the authors wish to address them:

1. For parameters like waist circumference and systolic blood pressure, I believe it would be more appropriate to use z-scores for age and sex, particularly as there were significant differences in terms of age between clusters and there is an over-representation of male participants.
2. As mentioned on line 143/144 and seen in Table 1, the early-mild metabotype has a significantly higher percentage of patients with advanced fibrosis when compared to the cardiometabolic metabotype and about 50% of the percentage of advanced fibrosis seen in the inflammatory-fibrotic metabotype. Knowing that fibrosis is the driver of outcomes from a liver point of view and knowing that this group was actually younger in age than the other 2 groups, makes it more worrying that this group potentially has more aggressive disease, rather than mild disease. How would the authors comment?
3. As mentioned on line 193/194 and Figure S5, tryptophan metabolism is enriched mostly in the inflammatory-fibrotic group,

then in the cardiometabolic group and then in the early-mild metabotype (albeit smaller differences between the CM and the EM group in comparison to the IF group). Knowing the association of tryptophan metabolism and fibrosis stage how would the authors reconcile that the fibrosis patterns don't quite follow this pattern amongst the groups (i.e. fibrosis was more in the IF>EM>CM)?

4. How do the authors believe that the exclusion of diabetic children from the study impact on the results?

5. Would it be possible, to have some suggested way from the authors of cluster prediction based on some of the clinical parameters? In other words, when presented with a patient how can we know which metabotype they potentially fit in?

Methodology is clear and presented in detail.

Overall a very interesting and useful study.

(Remarks on code availability)

N/A

Version 1:

Reviewer comments:

Reviewer #1

(Remarks to the Author)

The paper investigates the Metabolic Dysfunction-Associated Steatotic Liver Disease (MASLD) in 3 pediatric cohorts. In the review round, the authors addressed the questions about the data analysis, mostly in the domain of potential bias introduced by variable pre-selection. The authors conducted new experiments that include more variables and a different method benchmark, updated relevant text and figures. Overall, the comments are addressed and clarity improved.

(Remarks on code availability)

The code is provided alongside with installation instructions.

The reviewer tested one file and it was executing as expected

Reviewer #2

(Remarks to the Author)

Thank you for addressing the previous comments and revising the manuscript. The updated version is significantly improved. However, I remain concerned about the pathway analysis, which plays a central role in the biological interpretation of the metabolomics data in this manuscript.

While pathway analysis can offer useful insights into metabolic alterations, it must be applied rigorously. In this manuscript, the approach appears to rely on publicly available tools that, while convenient, often lack transparency and reproducibility, and may yield misleading results without careful validation.

For example, in Figure 2, "Tryptophan metabolism" is listed as a top pathway (6/41), and "Purine metabolism" as another (4/70). However, what do these ratios signify? Does "6/41" mean that only 6 of the 41 known metabolites in the tryptophan pathway were statistically significant? If so, that would be a relatively weak signal and may not justify emphasizing the pathway as enriched. The criteria used for pathway selection — including enrichment metrics, statistical thresholds, and background sets — are not clearly described and need to be explicitly stated.

Similarly, in Figure 4A, the metabolite PA (20:4) is highlighted. What is this molecule? Is it phospholipids???? It does not appear among the directly annotated metabolites and seems to be inferred only through the pathway tool. This raises questions about annotation confidence and interpretability.

The authors have really nice data. Using appropriate statistical analysis with correctly annotated metabolites, they can assess the biological importance without introducing any bias.

(Remarks on code availability)

Reviewer #3

(Remarks to the Author)

Thank you for the revision submitted.

I am happy with the response to my comments.

(Remarks on code availability)

Version 2:

Reviewer comments:

Reviewer #2

(Remarks to the Author)

The authors acknowledged the exploratory nature of pathways analysis and included the description of the methodology used. I have no further comments.

(Remarks on code availability)

REVIEWER COMMENTS – NATURE COMMUNICATIONS

Manuscript Title: *Clinically distinct metabotypes of pediatric MASLD identified through unsupervised clustering of NASH CRN data*

Manuscript ID: NCOMMS-25-11773-T

Corresponding Author: Helaina Huneault, PhD, RD

Date: June 30, 2025

Dear *Nature Communications* Editorial Team and Reviewers,

Thank you for the careful evaluation of our manuscript and the constructive feedback. We have thoroughly addressed all comments and revised the manuscript accordingly.

All changes in the revised manuscript are highlighted and indicated in bold. We have also completed the required checklists and other materials in accordance with *Nature Communications* policies.

We look forward to your further consideration of our revised submission.

Sincerely,

Helaina Huneault, PhD, RD (on behalf of the research team)

Reviewer #1 (Remarks to the Author):

The study investigates the Metabolic Dysfunction-Associated Steatotic Liver Disease (MASLD) in 3 pediatric cohorts assembled into one dataset containing demographic data, clinical targeted measurements, and untargeted metabolomics features. The authors present the analytical chemistry method for metabolic feature discovery and the data analysis pipeline. In the first step of the data analysis, clinically distinct metabotypes were formed with unsupervised clustering based on the 10 manually selected features. Following that, the comparative analysis of metabolomics data was performed to identify the most distinct metabolic features of each cluster. Some features were annotated and used in pathway enrichment analysis.

The authors present important work on pediatric cohorts in a clear and comprehensive way. The reviewer is not a wet lab specialist and therefore cannot comment on the specifics of the LC-MS method. Thus, the comments are focused on the data analysis methods.

1. The central claim of the paper is the existence of the three distinct metabotypes, which directs the follow-up statistical and pathway enrichment analysis. This analysis is performed based on the 10 preselected variables. While the unsupervised methods used for clustering and arriving at the conclusion about the existence of exactly 3 metabotypes are solid, the reviewer has concerns about the possible bias introduced by the choice of

variables. In the Discussion section, the authors make connections between the findings of the presented study and the existing literature. However, the 10 variables are also preselected on the basis of “prior use in adult MASLD phenotyping studies”, which makes them more likely to return similar results to the existing studies. The reviewer suspects that the results of the unsupervised clustering might significantly change when all the clinical and demographic variables are included. The results of the clustering that are based on all the variables must be included and compared to the presented metabotypes. Otherwise, the more detailed justification for not including the variables must be described.

We thank the reviewer for these thoughtful comments. We agree that variable selection can influence clustering results and appreciate the opportunity to clarify our approach. As described in our Methods section, we intentionally selected 10 clinically accessible variables based on their prior use in adult MASLD phenotyping studies, relevance to MASLD pathophysiology, and availability in routine clinical care. (**Methods section, Page 16, Lines 433-437**) This approach ensured that the identified metabotypes would be clinically interpretable, reproducible, and comparable to prior MASLD phenotyping studies.

The importance of consistent variable selection is well demonstrated in diabetes sub-phenotyping research. Since the initial study by Ahlqvist et al.¹, researchers have consistently used the same set of variables to perform clustering across different populations. This approach has led to novel insights; for example, in European populations, clusters tend to be driven by insulin resistance, whereas in South Asian populations, defects in insulin secretion are the predominant factors. Such comparisons reveal distinct underlying mechanisms in South Asians, which would have been difficult to identify/compare if different variable sets were used. Conversely, including a large number of additional variables can obscure biologically meaningful signals, introduce noise, and reduce the clarity of clustering outcomes. This concern is especially relevant for k-means clustering, where the Euclidean distance metric is sensitive to the number of variables; as this number increases, the meaning of distance changes, often leading to fewer and less distinct clusters².

As suggested by the reviewer, we conducted an exploratory clustering analysis using all available continuous clinical and demographic variables (n=21), including markers such as GGT, HbA1c, creatinine, bilirubin, and additional lipid measures. As shown below, the NbClust R package identified k=3 (a consensus of 26 metrics), and the silhouette method suggested k = 2 clusters. Despite this variability, the 3-cluster solution aligned most closely with the clinical phenotypes observed in our primary 10-variable model. Inclusion of additional variables led to minor changes in cluster membership but did not improve interpretability or alter the overall clustering structure.

NbClust (consensus of 26 clustering metrics):

Silhouette method:

Based on these findings, we retained the 10-variable clustering model to preserve clinical relevance, improve interpretability, and support comparability with prior MASLD phenotyping studies.

2. Do the same metatypes also occur when the unsupervised clustering of untargeted metabolomics features is performed? Do some other clusters form if only the metabolomics is measured and clinical variables are not included? How do they relate to the metatypes formed by the clinical variables and the previous studies?

We thank the reviewer for these important questions. In this study, clustering was intentionally performed using clinical variables to identify clinically meaningful and translatable pediatric MASLD phenotypes. Untargeted metabolomics was then used to explore metabolic differences

and compare these clinically defined metabolotypes. This two-step framework is now clarified in our updated Figure 6 (shown below). Additionally, we updated our methods section to highlight this: **Methods section, Pages 13, Lines 365-369: “This retrospective cross-sectional study employed a two-step framework (Figure 6). First, a data-driven clustering analysis was performed using anthropometric and clinical data from youth enrolled in NASH CRN studies to identify metabolotypes of pediatric MASLD. Second, an exploratory high-resolution metabolomics analysis of participant serum samples was conducted to examine differences in the metabolome among the identified metabolotypes.”** We did not perform unsupervised clustering based solely on untargeted metabolomics features, as this was beyond the scope of our analysis.

Figure 6. Two-step framework for identifying metabolotypes and exploring metabolic differences in pediatric MASLD. (1) Unsupervised clustering of clinical variables from 514 children enrolled in NASH CRN studies with biopsy-confirmed MASLD identified distinct metabolotypes using k-means clustering. (2) High-resolution metabolomics was performed on stored serum samples, including feature extraction (apLCMS, xMSanalyzer), normalization, and log₂ transformation. Features that differed significantly across metabolotypes ($p < 0.05$, FDR-adjusted) were identified using one-way ANOVA, entered into untargeted pathway analysis (Mummichog v2.0), and annotated using an internal reference library (tier 1) and xMSannotator. Metabolomics features were integrated with clinical biomarkers and fibrosis stage in xMWAS for network and pathway analysis. Features differing in centrality scores by ≥ 0.025 between metabolotype networks were further explored in pathway enrichment analysis.

While we did not cluster on metabolomics data alone, our study did examine how untargeted metabolomics features differed across the three clinically defined metabolotypes. This integrative approach allowed us to explore underlying biological differences, such as alterations in tryptophan metabolism, branched-chain amino acid degradation, and purine metabolism, which provided mechanistic insights into the distinct clinical profiles of the metabolotypes (**Discussion section, Pages 9-11, Lines 272-324**). We agree that future studies could explore clustering directly on metabolomics data; however, such analyses would benefit from advanced statistical methods and integration with clinical variables to ensure biological and clinical relevance. Furthermore, we have discussed how the metabolotypes formed in our analysis relate to previously reported subtypes in our **Discussion section, Pages 8-9, Lines 243-271**)

3. No participants from DB1 ended up in the inflammatory-fibrotic metabolotype. Do the authors suspect that this is due to chance or due to the selection bias of the participants in DB1?

We appreciate this thoughtful observation. We suspect that the absence of DB1 participants in the inflammatory-fibrotic metabolotype is primarily due to sampling variability, given the small number of DB1 participants included in the present analysis (n=23 with complete clinical and metabolomics data). With such a limited sample size, it is likely that the distribution across the three identified metabolotypes did not fully capture the heterogeneity present in the broader DB1 cohort. While we cannot entirely rule out the possibility of selection bias, we believe the lack of DB1 representation in the inflammatory-fibrotic group is more likely due to sample size constraints than to a systematic bias in participant selection.

4. To make an analysis on the Figure 4 even more explainable, the reviewer suggests including some potential cofactors (age, clinical variables) into a linear regression alongside with the most correlated metabolic features to predict the levels of fibrosis. This would allow to list out some possible cofactors and establish associations in the datasets.

We thank the reviewer for this suggestion. While we agree that incorporating clinical covariates into a regression model alongside selected metabolite features to predict fibrosis severity could provide additional insights, the primary focus of this study was to define clinically relevant metabolotypes of pediatric MASLD and characterize their distinct metabolomic profiles. Our goal was not to identify individual metabolites associated with fibrosis stage or to build a predictive model for fibrosis, as this represents a separate research question beyond the scope of the current analysis. We acknowledge that modeling fibrosis by integrating clinical and metabolomics data is an important future direction, and we have added this to our Discussion section as an avenue for future research in pediatric MASLD. **Discussion section, Page 13, Lines 354-356: “Moreover, further research could explore the relationship between specific metabolite features, clinical variables, and fibrosis severity in pediatric MASLD to identify potential biomarkers of fibrosis risk and refine our understanding of disease progression.”**

5. The reviewer suggests that the Supplementary Tables are provided in the Excel/CSV format for easy access.

We appreciate the reviewer's suggestion. We have now provided all Supplementary Tables in Excel format to facilitate ease of access.

Reviewer #1 (Remarks on code availability):

The code is provided alongside with installation instructions.
The reviewer tested one file and it was executing as expected.

Reviewer #2 (Remarks to the Author):

1. In this study the authors used a very well characterized cohort of 514 children with biopsy-proven MASLD to identify clinically distinct metabolic phenotypes of paediatric MASLD using unsupervised clustering and high-resolution metabolomics. This work with the aim to uncover distinct metabolic phenotypes in children with MASLD is very timely and important and a significant step in paediatric liver disease. In the first step, they used clinical and histological parameters and identified 3 unique clusters. This part is done very well. Although validation using external cohort is required. The second part related to metabolomics is more concerning. 3 cohorts were done in two groups possibly in two different labs and 2 years apart, hence the initial number of features are quite different and then no randomization in analysis which is quite important. The methodology is also less suitable for analysing lipids which are important in MASH and important potential biomarkers were reported previously in adult MASH patients. In addition, the results are not well-described. They reported 284 significant features but it is not clear if these are the metabolites that differentiate the 3 identified clusters or did they performed unsupervised analysis independently? Are these fully annotated using standards (tier 1) or these are only features or mixed? Then without discussing specificity of these metabolites, they moved on with performing pathway analysis using Mummichog or MetaboAnalyst which are ok for quick visualization but not reliable for such an important differentiation. I really don't think this type of analysis is relevant for this work and it offers any meaningful outcome. I suggest to remove it! This study has a lot of potential with a very important cohort. It would be great to use the same thorough evaluation of the metabolites as it was done for the clinical and histological parameters.

We thank the reviewer for recognizing the strengths of our clinical and histological analyses and for their detailed critique of the metabolomics component. We appreciate the opportunity to clarify and strengthen this aspect of the study and have made several key revisions and additions as detailed below.

We agree that validation in an external cohort is an important next step, and we have noted this as a limitation in the **Discussion section, Page 11-12, Lines 326- 329.**

We clarify that while samples from the DB2 cohort were processed in 2020 and those from the TONIC and DB1 cohorts in 2022, all analyses were conducted at the Emory Clinical Biomarkers Laboratory using consistent instrumentation, protocols, and internal quality control procedures. This minimized potential batch-related variability. To clarify this, we have added the following sentence to our **Methods section (Page 17, Lines 468-474):** **“While serum samples from the DB2 cohort were processed in 2020 and those from the TONIC and DB1 cohorts in 2022, all analyses were performed at the Emory Clinical Biomarkers Laboratory using established liquid chromatography-mass spectrometry (LC-MS) methods including identical instrumentation, protocols, and internal quality control procedures to minimize potential batch-related variability.”**

Additionally, as indicated in the **Methods section (Page 18, Lines 493-497):** **“The two separate HILIC+ and C18- datasets were matched and merged based on a m/z cutoff of 5 ppm and an RT cutoff of 30 seconds using R Studio software version 4.2.3. The resulting 3758 m/z -matched features from HILIC+ and 3520 from C18- were standardized across the studies, quantile normalized, and log₂-transformed using the online software MetaboAnalyst version 6.0 prior to downstream analysis.”** Furthermore, we clarify that all serum samples were randomized within each batch before extraction and LC-MS acquisition to reduce potential bias from instrument drift and batch effects (**Methods, Page 17, Lines 473-474**). **“Additionally, serum samples were randomized prior to extraction and LC-MS acquisition to minimize potential bias from instrument drift and batch position effects.”**

We have also updated Figure 6 to outline the two-step analysis framework used in this study (see response to reviewer #1, comment #2 above). Unsupervised clustering was first performed solely on clinical variables, followed by metabolomics analysis, which was conducted as a separate, exploratory analysis to characterize metabolic differences between these predefined metabolotypes.

Moreover, we have improved transparency regarding metabolite annotation. Among the significant features identified through one-way ANOVA (284 features from HILIC+ and 178 from C18-), the top 25 have now been annotated as shown in **Supplemental Tables S2 and S3**. These features were either confirmed using an internal reference library and authentic standards (MSI Level 1) or computationally annotated (MSI Level 2 or lower) following the Metabolomics Standards Initiative guidelines as described in the updated table descriptions. The significant features identified through pathway analysis were also annotated using the same process as indicated in our updated Methods section. **Methods section (Page 19, Lines 509-520):** **“The top 25 differentiating HILIC+ and C18- features were annotated using an internal reference library, confirmed by matching ion dissociation patterns and elution times with authentic standards³. These matches were considered level 1 ‘confirmed’ per the Metabolomics**

Standards Initiative (MSI)⁴. The remaining *m/z* features were computationally annotated using xMSannotator, which performs accurate mass matching to common positive and negative mode adducts in HMDB with an *m/z* tolerance of ± 5 parts per million (ppm) and a retention time tolerance of 10 seconds⁵. Pathway enrichment was conducted with Mummichog version 2.0⁶ in MetaboAnalyst 6.0⁷, with features identified through one-way ANOVA with FDR-adjusted *p*-values below 0.05 to assess pathway-level differences. Group-wise distributions of significantly enriched features were examined to interpret pathway differences between groups and visualized using the R packages *pheatmap*⁸ for heatmaps and *ggplot2*⁹ for violin plots. Significant features identified through pathway analysis (*p*<0.05) were also annotated as described above.”

We acknowledge that the pathway enrichment analysis conducted using Mummichog and MetaboAnalyst is exploratory and intended to provide initial insights into potential biological differences across metabolotypes, rather than definitive evidence of specific pathways or biomarkers. We have added a clarification to the **Methods section (Page 19, Lines 521-523)** to reflect this, stating: **“Pathway enrichment analysis was performed as an exploratory approach to investigate potential biological differences between the clinically defined metabolotypes, recognizing that further targeted studies are necessary to confirm these findings.”** In addition, the following statement has been added to the **Discussion/Limitations section (Page 13, Lines 357-358)**: **“Finally, our pathway enrichment analysis was exploratory and not intended to establish definitive pathways or biomarkers; targeted studies are required to validate these findings.”**

We also acknowledge the limitations of our untargeted metabolomics platform for analyzing lipids. Several lipid-related features, including phosphatidylcholine and sphingolipid species, were differentially abundant across metabolotypes. While our dual LC-MS platform (HILIC+ and C18-) approach offers broad coverage of the metabolome, it is not optimized for targeted lipidomics. To address this, we have added the following sentence to the **Discussion (Page 12-13, Lines 351-354)**: **“While our untargeted metabolomics approach offers broad coverage of the metabolome, it is not optimized for targeted lipidomics. Future studies employing dedicated lipidomics platforms are needed to better characterize lipid-driven mechanisms underlying pediatric MASLD heterogeneity.”**

We believe these substantial revisions, including the updated Figure 6 workflow, the clarified analysis strategy, the annotation of the top 25 features, and the expanded transparency around methods and limitations, address the reviewer’s concerns and significantly strengthen the metabolomics component of our study. We thank the reviewer for their constructive feedback, which has improved the quality and clarity of our manuscript.

2. Minor comment: Please use the recently approved term MASH/MASLD across the manuscript. Currently both MASH and NASH/NAFLD is used across the text.

Thank you for this helpful suggestion. We have carefully revised the manuscript to use the updated terminology MASH (metabolic dysfunction-associated steatohepatitis) and MASLD (metabolic dysfunction-associated steatotic liver disease) throughout, in accordance with the recently approved nomenclature. References to NASH/NAFLD have been retained only when referring to historical study names (e.g., NASH CRN) or previously published data that used the former terminology. These instances are now clarified in the text to avoid confusion.

Reviewer #3 (Remarks to the Author):

Thank you for asking me to review this manuscript. This was a very interesting to read, and relevant to the field, piece of work.

Noteworthy results:

The work presented looks to identify distinct phenotypes/metabotypes of paediatric MASLD by analysing clinical and metabolomic data from children with biopsy proven MASLD who had previously participated in 3 NASH CRN studies. (The inclusion of biopsy proven MASLD is a strength of this work).

With the use of unsupervised clustering, 3 distinct metabotypes were identified: the early-mild, the cardiometabolic, and the inflammatory-fibrotic, with differences in clinical, laboratory, and metabolic parameters between the groups. In addition six metabolic pathways were shown to be significantly altered between the three metabotypes.

Significance of work for the field:

This work would be of significance in the field, as it is very important to understand the different 'types' of MASLD in children from a disease pathogenesis point of view, from a prognostic point of view and of course from a management/ treatment point of view. Even more so as the prevalence of paediatric MASLD is rapidly increasing world wide. The work compares well to the relevant literature in adults with MASLD and the authors analyse this adequately in the discussion.

The work does support the conclusions and claims. The authors do highlight the limitations of the study, mainly the lack of a control group and the predominance of Hispanic males in the participants. This does limit the generalizability of the conclusions, but the work still provides valuable insight that would be very helpful in the design of future studies, including longitudinal research.

In terms of data analysis, interpretation and conclusions, there are a few points that may strengthen the manuscript should the authors wish to address them:

1. For parameters like waist circumference and systolic blood pressure, I believe it would be more appropriate to use z-scores for age and sex, particularly as there were significant

differences in terms of age between clusters and there is an over-representation of male participants.

We thank the reviewer for this helpful suggestion. While age- and sex-adjusted percentiles or z-scores are commonly used in pediatric studies, their application in our MASLD cohort is limited. The 2017 American Academy of Pediatrics guidelines provide SBP percentiles based on normal-weight children, which may not generalize to youth with overweight or obesity (Flynn et al., 2017)¹⁰. Similarly, the z-score equations from Wühl et al. (2002)¹¹ were derived from normal-weight European children and may not apply to diverse or overweight populations. For WC, no widely accepted z-score references exist across the full pediatric age and BMI spectrum (Yamanaka et al., 2021; Fernández et al., 2004)^{12,13}.

In our study, age was included as a clustering variable, and all features, including SBP and WC, were standardized prior to clustering to ensure equal weighting, as described in the **Methods (Page 16, Lines 437–439): “All variables were scaled to a mean of 0 and a standard deviation of 1 before performing k-means clustering to ensure comparability across different scales.”** This approach maintains clinical interpretability and aligns with prior MASLD phenotyping studies. While male participants were overrepresented, sex was not a primary driver of cluster separation based on principal component analysis (**Figure S2**).

2. As mentioned on line 143/144 and seen in Table 1, the early-mild metabotype has a significantly higher percentage of patients with advanced fibrosis when compared to the cardiometabolic metabotype and about 50% of the percentage of advanced fibrosis seen in the inflammatory-fibrotic metabotype. Knowing that fibrosis is the driver of outcomes from a liver point of view and knowing that this group was actually younger in age than the other 2 groups, makes it more worrying that this group potentially has more aggressive disease, rather than mild disease. How would the authors comment?

We thank the reviewer for raising this important observation. We hypothesize that the higher proportion of advanced fibrosis in the early-mild metabotype, despite its younger age profile, may reflect a mixed, earlier-stage phenotype that shares features of both the cardiometabolic and inflammatory-fibrotic groups. This group may include individuals who could progress toward either of these more defined phenotypes over time (**Figure 5**).

Additionally, while the early-mild group shows fewer cardiometabolic risk factors overall, it is important to note that fibrosis can develop early in life, even in the absence of overt metabolic dysfunction. The presence of advanced fibrosis in this younger group may therefore indicate an underlying vulnerability or disease trajectory that warrants close monitoring.

We have added this point to the Discussion to highlight the need for future longitudinal studies to better understand progression pathways and the clinical significance of the early-mild group. **Discussion section, Page 8, Lines 228-230: “As illustrated in Figure 5, we hypothesize that**

the early-mild metabotype represents a younger, less severe phenotype with mixed features of the other two metabotypes, which may progress into one of them over time.” Also, Page 8, Lines 239-242: “The higher proportion of advanced fibrosis observed in the early-mild group underscores the potential for fibrosis to develop early in life, even in the absence of overt metabolic dysfunction, and highlights the need for longitudinal studies to better understand progression pathways in pediatric MASLD¹⁴.”

3. As mentioned on line 193/194 and Figure S5, tryptophan metabolism is enriched mostly in the inflammatory-fibrotic group, then in the cardiometabolic group and then in the early-mild metabotype (albeit smaller differences between the CM and the EM group in comparison to the IF group). Knowing the association of tryptophan metabolism and fibrosis stage how would the authors reconcile that the fibrosis patterns don't quite follow this pattern amongst the groups (i.e. fibrosis was more in the IF>EM>CM)?

We thank the reviewer for this thoughtful observation. While tryptophan metabolism was most enriched in the inflammatory-fibrotic group, followed by the cardiometabolic and early-mild groups, the fibrosis distribution followed a different pattern (IF > EM > CM). We hypothesize that this reflects the complexity of tryptophan metabolism, which is involved in multiple biological processes beyond fibrosis, including obesity-associated low-grade chronic inflammation, metabolic dysfunction, and immune regulation. Therefore, while tryptophan pathway enrichment is highest in the inflammatory-fibrotic group, its presence in the cardiometabolic group may reflect metabolic inflammation rather than fibrosis per se.

We have added this interpretation to the Discussion to emphasize the need for future targeted studies to clarify the specific roles of tryptophan metabolism in pediatric MASLD progression. **Discussion section, Page 10, Lines 284-290: “Tryptophan catabolism is also upregulated in obesity and is associated with systemic low-grade chronic inflammation^{15,16}. While enrichment of tryptophan metabolism in the inflammatory-fibrotic group likely reflects fibrosis-related processes, its elevated presence in the cardiometabolic group may reflect underlying metabolic inflammation. Fibrosis progression in pediatric MASLD likely results from the combined effects of altered tryptophan metabolism, lipid metabolism, and inflammation. Future studies are needed to clarify the specific contributions of tryptophan metabolism to fibrosis development and disease progression in this population.”**

4. How do the authors believe that the exclusion of diabetic children from the study impact on the results?

We appreciate the reviewer’s thoughtful question. As shown in the participant selection workflow (**Supplementary Figure S14**), participants with diabetes (HbA1c \geq 6.5%) were not explicitly excluded during initial data selection. However, because k-means clustering is

sensitive to extreme values, we applied an objective outlier removal process, excluding participants with any clinical variable greater than five standard deviations from the mean. This approach resulted in the exclusion of all but one participant with an HbA1c $\geq 6.5\%$. To maintain consistency and avoid undue influence from a single extreme value, we also excluded this remaining participant prior to analysis. This exclusion slightly reduced the sample size but improved the robustness of the clustering by minimizing the influence of extreme outliers.

We have clarified this step in the **Methods section (Page 14, Lines 390-397)**: "**The initial eligible sample included 556 children and adolescents with available clinical and metabolomics data. A small percentage of missing data (0.34%) was imputed using the median for clinical variables. Outliers, defined as participants with any clinical variable greater than five standard deviations from the mean, were excluded, resulting in 515 participants. During this process, most participants with an HbA1c $\geq 6.5\%$ were excluded; however, one participant remained in the dataset. To maintain consistency and avoid bias from a single extreme value, this participant was also excluded, yielding a final sample of 514 children and adolescents for analysis (Figure S14).**"

5. Would it be possible, to have some suggested way from the authors of cluster prediction based on some of the clinical parameters? In other words, when presented with a patient how can we know which metabotype they potentially fit in?

Thank you for this insightful suggestion. Developing a clinical tool for predicting metabotype membership is an important focus of our ongoing work. Similar to the approach by Raverdy et al. (Nature Medicine, 2024)¹⁷, we plan to create a calculator that will allow clinicians to input a patient's values for the 10 clinical variables used in clustering and compute the Euclidean distance to each cluster centroid. The patient would then be assigned to the metabotype with the closest centroid. However, we recognize that this approach requires careful validation, and our first priority is to validate the metatypes identified in this study in an independent cohort. Once validated, we will refine and test the calculator, which we plan to include in a future publication.

Methodology is clear and presented in detail.
Overall a very interesting and useful study.

Reviewer #3 (Remarks on code availability):
N/A

References

1. Ahlqvist E, Storm P, Käräjämäki A, et al. Novel subgroups of adult-onset diabetes and their association with outcomes: a data-driven cluster analysis of six variables. *Lancet Diabetes Endocrinol*. May 2018;6(5):361-369. doi:10.1016/s2213-8587(18)30051-2
2. Steinley D, Brusco MJ. Choosing the number of clusters in K-means clustering. *Psychol Methods*. Sep 2011;16(3):285-97. doi:10.1037/a0023346
3. Go Y-M, Walker DI, Liang Y, et al. Reference Standardization for Mass Spectrometry and High-resolution Metabolomics Applications to Exposome Research. *Toxicological Sciences*. 2015;148(2):531-543. doi:10.1093/toxsci/kfv198
4. Sumner LW, Amberg A, Barrett D, et al. Proposed minimum reporting standards for chemical analysis Chemical Analysis Working Group (CAWG) Metabolomics Standards Initiative (MSI). *Metabolomics*. Sep 2007;3(3):211-221. doi:10.1007/s11306-007-0082-2
5. Uppal K, Walker DI, Jones DP. xMSannotator: An R Package for Network-Based Annotation of High-Resolution Metabolomics Data. *Anal Chem*. Jan 17 2017;89(2):1063-1067. doi:10.1021/acs.analchem.6b01214
6. Li S, Park Y, Duraisingham S, et al. Predicting network activity from high throughput metabolomics. *PLoS Comput Biol*. 2013;9(7):e1003123. doi:10.1371/journal.pcbi.1003123
7. Pang Z, Lu Y, Zhou G, et al. MetaboAnalyst 6.0: towards a unified platform for metabolomics data processing, analysis and interpretation. *Nucleic Acids Res*. Jul 5 2024;52(W1):W398-w406. doi:10.1093/nar/gkae253
8. Kolde R, Kolde MR. Package 'pheatmap'. *R package*. 2015;1(7):790.
9. Wickham H. ggplot2. *Wiley interdisciplinary reviews: computational statistics*. 2011;3(2):180-185.
10. Flynn JT, Kaelber DC, Baker-Smith CM, et al. Clinical Practice Guideline for Screening and Management of High Blood Pressure in Children and Adolescents. *Pediatrics*. Sep 2017;140(3)doi:10.1542/peds.2017-1904
11. Wühl E, Witte K, Soergel M, Mehls O, Schaefer F. Distribution of 24-h ambulatory blood pressure in children: normalized reference values and role of body dimensions. *J Hypertens*. Oct 2002;20(10):1995-2007. doi:10.1097/00004872-200210000-00019
12. Yamanaka AB, Davis JD, Wilkens LR, et al. Determination of Child Waist Circumference Cut Points for Metabolic Risk Based on Acanthosis Nigricans, the Children's Healthy Living Program. *Prev Chronic Dis*. Jun 24 2021;18:E64. doi:10.5888/pcd18.210021
13. Fernández JR, Redden DT, Pietrobelli A, Allison DB. Waist circumference percentiles in nationally representative samples of African-American, European-American, and Mexican-American children and adolescents. *J Pediatr*. Oct 2004;145(4):439-44. doi:10.1016/j.jpeds.2004.06.044
14. Schwimmer JB, Behling C, Newbury R, et al. Histopathology of pediatric nonalcoholic fatty liver disease. *Hepatology*. Sep 2005;42(3):641-9. doi:10.1002/hep.20842
15. Oxenkrug GF. Tryptophan kynurenine metabolism as a common mediator of genetic and environmental impacts in major depressive disorder: the serotonin hypothesis revisited 40 years later. *Isr J Psychiatry Relat Sci*. 2010;47(1):56-63.

16. Mangge H, Stelzer I, Reininghaus EZ, Weghuber D, Postolache TT, Fuchs D. Disturbed tryptophan metabolism in cardiovascular disease. *Curr Med Chem*. Jun 2014;21(17):1931-7. doi:10.2174/0929867321666140304105526
17. Raverdy V, Tavaglione F, Chatelain E, et al. Data-driven cluster analysis identifies distinct types of metabolic dysfunction-associated steatotic liver disease. *Nature Medicine*. 2024/12/09 2024;doi:10.1038/s41591-024-03283-1

REVIEWER COMMENTS – NATURE COMMUNICATIONS

Manuscript Title: *Clinically distinct metabotypes of pediatric MASLD identified through unsupervised clustering of NASH CRN data*

Manuscript ID: NCOMMS-25-11773A

Corresponding Author: Helaina Huneault, PhD, RD

Date: August 28, 2025

Dear *Nature Communications* Editorial Team and Reviewers,

Thank you for your thoughtful feedback and the opportunity to revise our manuscript. We have carefully addressed all remaining reviewer comments, with particular attention to clarifying our pathway analysis and annotation methods. Below, we provide a detailed point-by-point response. All changes in the revised manuscript are highlighted and indicated in bold.

We appreciate your continued consideration.

Sincerely,

Helaina Huneault, PhD, RD (on behalf of the research team)

REVIEWER COMMENTS

Reviewer #1 (Remarks to the Author):

The paper investigates the Metabolic Dysfunction-Associated Steatotic Liver Disease (MASLD) in 3 pediatric cohorts. In the review round, the authors addressed the questions about the data analysis, mostly in the domain of potential bias introduced by variable pre-selection. The authors conducted new experiments that include more variables and a different method benchmark, updated relevant text and figures. Overall, the comments are addressed and clarity improved.

Reviewer #1 (Remarks on code availability):

The code is provided alongside with installation instructions.

The reviewer tested one file and it was executing as expected

Thank you for reviewing our manuscript and code.

Reviewer #2 (Remarks to the Author):

Thank you for addressing the previous comments and revising the manuscript. The updated version is significantly improved. However, I remain concerned about the pathway analysis,

which plays a central role in the biological interpretation of the metabolomics data in this manuscript.

While pathway analysis can offer useful insights into metabolic alterations, it must be applied rigorously. In this manuscript, the approach appears to rely on publicly available tools that, while convenient, often lack transparency and reproducibility, and may yield misleading results without careful validation.

For example, in Figure 2, "Tryptophan metabolism" is listed as a top pathway (6/41), and "Purine metabolism" as another (4/70). However, what do these ratios signify? Does "6/41" mean that only 6 of the 41 known metabolites in the tryptophan pathway were statistically significant? If so, that would be a relatively weak signal and may not justify emphasizing the pathway as enriched. The criteria used for pathway selection — including enrichment metrics, statistical thresholds, and background sets — are not clearly described and need to be explicitly stated.

Similarly, in Figure 4A, the metabolite PA (20:4) is highlighted. What is this molecule? Is it phospholipids???? It does not appear among the directly annotated metabolites and seems to be inferred only through the pathway tool. This raises questions about annotation confidence and interpretability.

The authors have really nice data. Using appropriate statistical analysis with correctly annotated metabolites, they can assess the biological importance without introducing any bias.

We thank the reviewer for their careful assessment and constructive feedback regarding pathway enrichment analysis and metabolite annotation. We have taken several steps to improve the transparency of our methods and clarify the interpretation of our results.

We agree that the ratios in **Figure 2** were not clearly explained in the initial submission. While this information was included in the figure legend, it was missing from the Methods. In the revised manuscript, we now explicitly state that the numbers in parentheses indicate the number of statistically significant metabolic features (numerator) out of the number of detected metabolic features within each pathway (denominator), as defined by the KEGG pathway library. In line with default parameters in Mummichog and MetaboAnalyst, we required at least three significant features per pathway to reduce the likelihood of spurious enrichment results.

We have also revised the Methods section to clearly describe the enrichment criteria, statistical thresholds, and background sets used for pathway analysis. **Methods section, Pages 19-20, Lines 527–542: “Pathway enrichment was conducted with Mummichog version 2.0¹ in MetaboAnalyst 6.0². Significant features identified through one-way ANOVA with FDR-adjusted p-values <0.05 were used as input. Enrichment was evaluated via permutation testing using all quality-filtered m/z features as the background. Pathways were considered enriched if the permutation-derived p-value was <0.05 and at least three significant features overlapped with known metabolites in the pathway. Pathway membership was defined according to the Kyoto Encyclopedia of Genes and Genomes (KEGG) library.**

Analyses were performed with the following parameters: mass tolerance 5 ppm; RT in seconds; primary ions enforced; and inclusion of pathways/metabolite sets containing ≥ 3 entries. Significant features identified through pathway analysis were also annotated as described above. The pathway results include ratios that indicate the number of significant features mapped to each pathway (numerator) out of the total number of pathway-relevant features (denominator). Group-wise distributions of the significant features were examined to interpret pathway differences between groups and visualized using the R packages *pheatmap*³ for heatmaps and *ggplot2*⁴ for violin plots. Pathway enrichment was performed as an exploratory approach to investigate potential biological differences between clinically defined metabolotypes, recognizing that further targeted studies will be necessary to confirm these findings.”

Additionally, we updated **Supplementary Table S4** to include the Schymanski Level Confidence (SLC) for each annotation (see Liu et al., 2020, *Analytical Chemistry*). Several of the significantly enriched pathways contained at least one Level 1 annotation from our internal reference library, providing strong confidence in these results. Three pathways, propanoate metabolism, BCAA degradation, and pantothenate and CoA biosynthesis, included features such as 3-methyl-2-oxobutanoic acid and 4-methyl-2-oxopentanoate that could not be corroborated by either xMSannotator or our internal reference library. These features were therefore conservatively reported as SLC Level 5. We have noted these as exploratory, hypothesis-generating findings that warrant confirmation in future targeted studies. The datasets used as input for pathway analysis are now provided in **Supplementary Tables S9 and S10**, and our Methods and Results sections have been updated accordingly.

Methods section, Page 19, Lines 542-543: “The datasets used as input for pathway analysis via MetaboAnalyst are provided in Supplementary Tables S9 and S10.”

Results section, Pages 5-6, Lines 174-180: “Among the significantly enriched pathways, several contained at least one Schymanski Level Confidence (SLC) Level 1 identification from our internal reference library, supporting the confidence of these results⁵. Three pathways, propanoate metabolism, BCAA degradation, and pantothenate and CoA biosynthesis, contained features that could not be independently confirmed by secondary annotation methods and were therefore reported as SLC Level 5. These results should be interpreted cautiously but may represent true biological signals and provide important hypotheses for future validation.”

We have also updated the explanation of our metabolite annotation procedures for clarity. **Methods section, Page 19, Lines 518–526: “...features were first annotated using an internal reference library and confirmed by matching ion dissociation patterns and retention times with authentic standards, which were reported as Schymanski Level 1⁶. The remaining features were computationally annotated using xMSannotator, which performs accurate mass matching to common positive and negative mode adducts in the Human**

Metabolome Database (HMDB) with an m/z tolerance of ± 5 parts per million (ppm) and a retention time tolerance of 10 seconds⁷. Annotation confidence was reported using the Schymanski Level Confidence (SLC) framework, which ranges from Level 1 (confirmed structure with reference standard) to Level 5 (exact mass of interest only)⁵.

Regarding **Figure 4A**, the feature initially described as PA (20:4) has been corrected to PA(O-22:4), consistent with the HMDB11156 candidate returned by xMSannotator. PA refers to phosphatidic acid, with O- denoting an ether-linked species. Because this annotation is based on accurate mass without MS/MS confirmation and without determination of an unequivocal molecular formula, the feature is conservatively reported as SLC Level 5. We clarify that all metabolite features in **Figure 4** were annotated using either our internal reference library or xMSannotator, as described above. This is now stated in our updated Methods.

Methods section, Page 21, Lines 571–574: “The top 20 significant feature correlations ($p < 0.05$) were annotated using an internal reference library or xMSannotator, as described above, and visualized in a Manhattan plot, with p -values transformed to negative \log_{10} to highlight significance levels.”

We have also updated Supplementary **Figure S8** to include the compound name, adduct, and SLC for each of the top 20 annotated features from HILIC+ and C18– shown in **Figure 4**, with the corrected annotation listed as PA(O-22:4), SLC Level 5. To improve interpretability, we added an abbreviation key to the legend of Figure 4: **“Figure 4. Top 20 HILIC+ (A) and C18– (B) metabolite features associated with fibrosis among NASH CRN study participants. Correlations are presented on a color scale from -1 (blue) to 1 (red). The dashed line indicates a p -value cutoff of <0.05 . Metabolite features were annotated using the internal reference library and xMSannotator. Please see Table S8 for correlations, p -values, adducts, and annotation confidence. Abbreviations: PA, phosphatidic acid; PC, phosphatidylcholine; PE, phosphatidylethanolamine; PI, phosphatidylinositol. O- denotes an ether-linked species.”**

Lastly, we have updated our Discussion section to emphasize the exploratory nature of the metabolomics analysis and findings. **Discussion section, Page 13, Lines 368-372: “Our findings underscore the potential for precision healthcare approaches to tailor interventions to specific metabolic dysfunctions in MASLD subtypes. Given the exploratory nature of the metabolomics analysis, further longitudinal and targeted studies are needed to validate these findings and explore strategies that may improve outcomes for children with MASLD.”**

We sincerely appreciate the reviewer’s comments, which helped us significantly strengthen the clarity and interpretability of our pathway analysis and feature annotation methods.

Reviewer #3 (Remarks to the Author):

Thank you for the revision submitted.
I am happy with the response to my comments.

Thank you for reviewing our manuscript.

References

1. Li S, Park Y, Duraisingham S, et al. Predicting network activity from high throughput metabolomics. *PLoS Comput Biol*. 2013;9(7):e1003123. doi:10.1371/journal.pcbi.1003123
2. Pang Z, Lu Y, Zhou G, et al. MetaboAnalyst 6.0: towards a unified platform for metabolomics data processing, analysis and interpretation. *Nucleic Acids Res*. Jul 5 2024;52(W1):W398-w406. doi:10.1093/nar/gkae253
3. Kolde R, Kolde MR. Package ‘pheatmap’. *R package*. 2015;1(7):790.
4. Wickham H. ggplot2. *Wiley interdisciplinary reviews: computational statistics*. 2011;3(2):180-185.
5. Liu KH, Nellis M, Uppal K, et al. Reference Standardization for Quantification and Harmonization of Large-Scale Metabolomics. *Anal Chem*. Jul 7 2020;92(13):8836-8844. doi:10.1021/acs.analchem.0c00338
6. Go Y-M, Walker DI, Liang Y, et al. Reference Standardization for Mass Spectrometry and High-resolution Metabolomics Applications to Exposome Research. *Toxicological Sciences*. 2015;148(2):531-543. doi:10.1093/toxsci/kfv198
7. Uppal K, Walker DI, Jones DP. xMSannotator: An R Package for Network-Based Annotation of High-Resolution Metabolomics Data. *Anal Chem*. Jan 17 2017;89(2):1063-1067. doi:10.1021/acs.analchem.6b01214

Responses to the Reviewers

Clinically distinct metabotypes of pediatric MASLD identified through unsupervised clustering of NASH CRN data

Nature Communications manuscript NCOMMS-25-11773B

December 3, 2025

Reviewer #2 (Remarks to the Author):

1) The authors acknowledged the exploratory nature of pathways analysis and included the description of the methodology used. I have no further comments.

We thank the reviewer for their feedback and are pleased that the clarification of the exploratory pathway analysis and methodological description addressed their concerns. No further changes were required in response to this comment.